# Structure-based design of a Cortistatin analogue with immunomodulatory activity in models of inflammatory bowel disease

Álvaro Rol[1], Toni Todorovski [1,6], Pau Martin-Malpartida [1], Anna Escolà[1], Elena Gonzalez-Rey[2], Eric Aragón[1], Xavier Verdaguer[1,3], Mariona Vallès-Miret [4], Josep Farrera-Sinfreu[4], Eduard Puig [1], Jimena Fernández-Carneado[4], Berta Ponsati[4], Mario Delgado [2✉], Antoni Riera [1,3✉] & Maria J. Macias [1,5✉]

Ulcerative colitis and Crohn's disease are forms of inflammatory bowel disease whose incidence and prevalence are increasing worldwide. These diseases lead to chronic inflammation of the gastrointestinal tract as a result of an abnormal response of the immune system. Recent studies positioned Cortistatin, which shows low stability in plasma, as a candidate for IBD treatment. Here, using NMR structural information, we design five Cortistatin analogues adopting selected native Cortistatin conformations in solution. One of them, A5, preserves the anti-inflammatory and immunomodulatory activities of Cortistatin in vitro and in mouse models of the disease. Additionally, A5 displays an increased half-life in serum and a unique receptor binding profile, thereby overcoming the limitations of the native Cortistatin as a therapeutic agent. This study provides an efficient approach to the rational design of Cortistatin analogues and opens up new possibilities for the treatment of patients that fail to respond to other therapies.

[1] Institute for Research in Biomedicine (IRB-Barcelona), The Barcelona Institute of Science and Technology, Barcelona, Spain. [2] Instituto de Parasitología y Biomedicina López-Neyra (IPBLN-CSIC), Armilla, Granada, Spain. [3] Departament de Química Inorgànica i Orgànica, Secció de Química Orgànica, Universitat de Barcelona, Barcelona, Spain. [4] BCN Peptides S.A. Pol.Ind. Els Vinyets-Els Fogars, Barcelona, Spain. [5] Institució Catalana de Recerca i Estudis Avançats (ICREA), Barcelona, Spain. [6] Present address: Departament de Ciències Experimentals i de la Salut, Universitat Pompeu Fabra, Barcelona, Spain. ✉email: mdelgado@ipb.csic.es; antoni.riera@irbbarcelona.org; maria.macias@irbbarcelona.org

The gastrointestinal tract is a complex system in which factors such as diet, environmental and/or microbial inputs continuously challenge the immune system[1]. When these factors dysregulate the gastrointestinal immune system – especially in genetically predisposed individuals – the perturbation gives rise to inflammation and other symptoms collectively known as inflammatory bowel disease (IBD). In fact, IBD covers a broad spectrum of pathologies and disease severities, including Ulcerative colitis (UC) and Crohn's disease (CD), both of which lead to chronic inflammations. Although the underlying mechanisms involved in these diseases are not fully understood, they correlate with elevated levels of pro-inflammatory cytokines, tumour necrosis factor-α (TNF-α), interferon-γ (IFNγ) and interleukin 2 (IL-2)[1,2]. In most instances, no cure is currently available, thus turning IBD into a chronic condition, which affects the quality of patients' life to different extents and represents a major health concern for society. Finding specific and safe therapeutic tools to tackle these complex disorders require an efficient combination of approaches to increase treatment effectiveness, improve patients' quality of life and reduce the economic cost for the public health system[3].

The current treatment pipeline includes amino-salicylates, corticosteroids, immunomodulators, anti-TNFα antibodies and surgery[4]. Anti-TNF agents have been a major advance as they have proven to be highly effective for some UC and CD patients. Unfortunately, a third of the patients does not respond to the treatment, thereby highlighting the need for alternative therapeutic strategies[5]. In this regard, the main challenges that patients face are disease progression, uncontrolled and relapsing inflammation, the appearance of important side-effects in long-term treatments and the high economic cost (over $20 billion every year in the USA)[6]. Therefore, the identification of other molecules to use in combination with or as alternatives to drugs currently available is a major need in the field.

Several years ago, the neuropeptide Cortistatin (CST) was found to play immunomodulatory roles[7–9] and its external administration protected mouse models of IBD from developing colitis[10]. CST mRNA was detected in many neuroendocrine tumours of the lung and in neuroendocrine tumours of the gastrointestinal tract[11,12]. CST sequences (human, mouse and rat) are highly similar to that of human Somatostatin (SST, Fig. 1a), another natural peptide that has a broad set of biological actions and has been extensively studied[13]. Remarkably, both CST and SST bind to the same set of SST-Receptors (SSTR1-5)[14]. CST, but not SST, inhibits the in vitro production of inflammatory mediators in activated macrophages, thereby revealing that it is an effective agent against hapten-induced CD[9,10]. These anti-inflammatory effects would be explained by the regulation of cytokines mediated by CST, decreasing the release of pro-inflammatory and cytotoxic cytokines such as TNFα, IFNγ, IL2, IL6 and nitric oxide, while stimulating the production of anti-inflammatory IL10. However, the short lifespan (two minutes in serum) of CST limits its potential application as a therapeutic agent for IBD. In fact, CST and SST share similar limitations with respect to stability, thereby precluding their direct pharmacological use. In the case of SST, these limitations have been overcome by designing SST analogues containing non-native amino acids and/or by reducing the length of the peptide. Some of these analogues, such as octreotide (Sandostatin®), lanreotide (Somatuline®), vapreotide (Sanvar®) and pasireotide (Signifor®) have reached the market.

We contemplated the possibility of improving the stability and immunoregulatory properties of CST by introducing non-native amino acids at specific positions – as previously done for SST – while maintaining the full-length peptide chain. To facilitate the rational design of the analogues, we first studied the conformations of the native neuropeptide in solution by nuclear magnetic resonance (NMR). Although the peptide is conformationally flexible, we observed different aromatic clusters in solution whose conformations were in equilibrium. We hypothesised that these distinct clusters correlate with the neuropeptide's function and we set to design analogues that could potentially populate some of these clusters as the main conformation in solution. Among these peptide analogues designed and tested, analogue 5 showed higher stability in serum while exerting the anti-inflammatory activity shown by the native peptide in two established preclinical models of IBD[15]. The activity of this analogue was of the same order of magnitude as that of the anti-TNFα antibody and Mesalazine (reference treatments), which were both used for comparison. Moreover, by solving the structure of this analogue, we have been able to correlate a specific aromatic cluster with the anti-inflammatory activity of CST.

Our work provides a framework to design novel compounds to treat IBD. These compounds have scaffolds based on the natural neuropeptide CST, thus maintaining its immunoregulatory activity while displaying improved stability and specificity with respect to the natural neuropeptide in models of IBD.

## Results

**Conformational properties of Cortistatin in solution by NMR.** In contrast to the wealth of conformational studies available for SST and its analogues[16–22], there is no information on the conformations of CST in solution to date. To bridge this gap of knowledge, we synthesised CST (r,mCST-14, Fig. 1a) using standard Fmoc/ᵗBu solid-phase peptide synthesis (as described in the Methods section and in Supplementary Fig. 1a)[23], and studied its folding properties in solution by NMR.

The 2D NMR spectra for this peptide allowed us to obtain the sequence specific resonance assignments[24], as well as torsion angles and inter-proton NOE distances (nuclear Overhauser effects) under these experimental conditions. These restraints were collected (experimental section) to calculate the structural ensemble in solution using the Crystallography & NMR system software suite (CNS)[25]. The pair of conserved cysteine residues enables the peptide to form an internal disulfide bond and a cyclic peptide structure with both the N- and C-termini on one side of the molecule (Fig. 1b). We also detected NOEs between the side chains of Trp7 and Lys8, denoting the presence of a type II β-turn at the other side of the disulfide bridge. These same residues and the turn are also observed in SST[26]. Moreover, weak NOEs were detected between protons of the phenylalanine rings, some of them incompatible with a unique spatial arrangement, thereby confirming the presence of different aromatic clusters in solution. Indeed, the ensemble of CST contains subfamilies of conformers defined mainly by the presence of aromatic clusters, where either two or three aromatic Phe residues pack together (Fig. 1b). Overall, these results suggest that different sets of conformations can be adopted by the native neuropeptide in solution, and in a context-dependent environment, these different orientations might be specifically selected by distinct protein/receptor partners in order to regulate cellular responses.

**De novo design of Cortistatin analogues.** Encouraged by these results, we designed CST-based analogues, envisioning that they could individually populate – like the main conformation – each of these aromatic clusters (Fig. 1c, analogues 2–6). We also aimed to correlate the conformations present in the native CST ensemble and in the analogues, with a given observable in our functional assays, enabling the future design of molecules with immunoregulatory activity and improved stability.

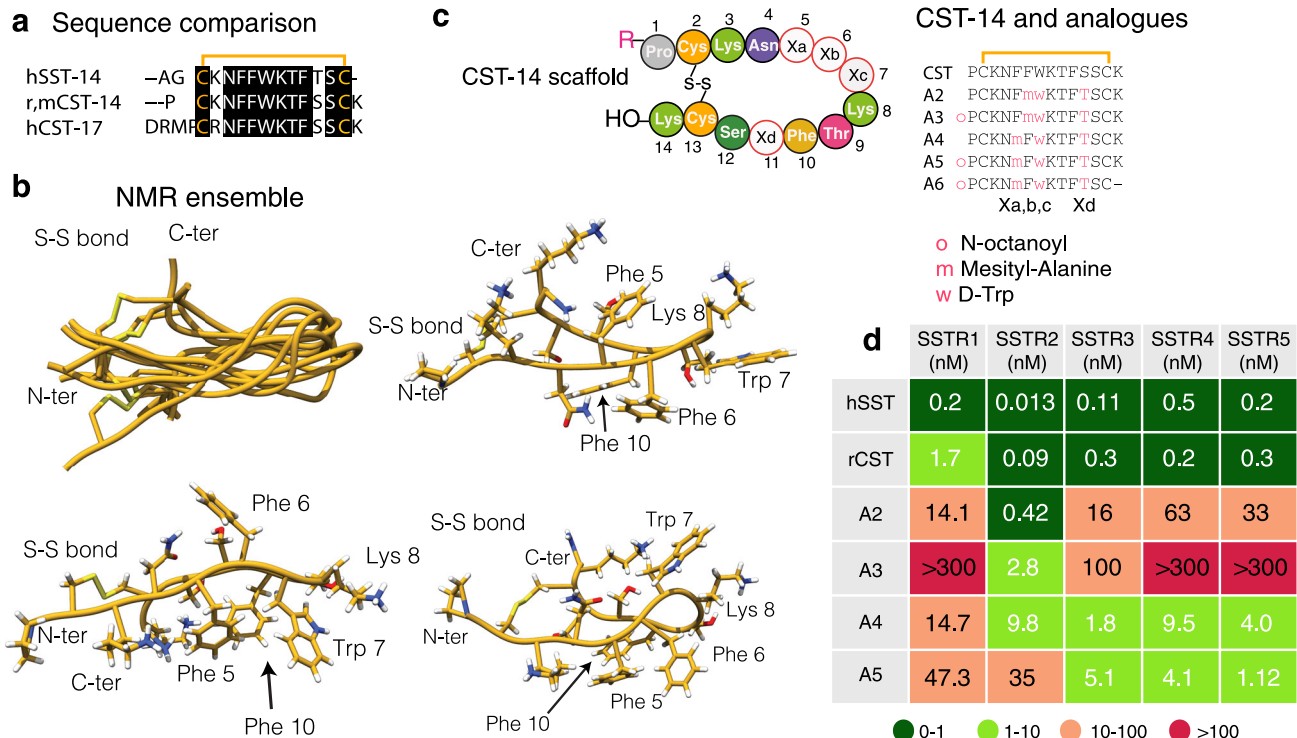

**Fig. 1 Cortistatin and Somatostatin sequence comparison, analogue design and NMR conformers. a** Sequence comparison of human Somatostatin (SST) to rat, mouse and human Cortistatins (CST). **b** Backbone superposition of the 10-lowest energy conformers. The peptide backbone is shown in gold and the N- and C-terminus residues, as well as the disulfide bond are labelled. Representative aromatic clusters. All side chains are shown to facilitate structural analysis. Side chains participating in the interactions are specifically labelled. **c** Schematic representation of Cortistatin-14 sequence. Positions Xa, Xb, Xc and Xd have been mutated to prepare the analogues. Native CST was also synthesised for the NMR and functional studies. Sequences of the analogues prepared in this work. Substitutions are coloured. Synthetic scheme is displayed as Supplementary Fig. 1a. **d** Schematic representations of the relative affinities of SST, CST and analogues A2-A5 for the different Somatostatin receptors (SSTR1-SSTR5) Supplementary Fig. 1b. The relative affinities are represented based on the determined $IC_{50}$ values. The values are collected as Supplementary Table 1.

To design the first CST analogues, we collected the information described in the literature to optimise the stability and properties of SST. For instance, it is well known that substitution of L-Trp to D-Trp increases the population of the native β-turn and of and the analogue half-life in serum[27]. Also, substitutions of Phe by L-mesityl alanine (2,4,6-trimethylphenylalanine, Msa) favour π–π interactions between aromatic rings and increase the overall conformational stability of SST derivatives[19,20]. For these reasons, in all the CST analogues, L-Trp was replaced by D-Trp in position 7 and Ser11 was replaced by Thr11 to stabilise the β-turn; and either Phe6 or Phe5 were replaced by Msa (analogues **2** and **4**; Fig. 1c). In addition to these modifications, terminal residues were acylated as N-octanoyl amides (analogues **3** and **5**). The reasoning behind this modification was to increase the hydrophobicity of the analogue and mimic a similar modification to that observed in ghrelin, a neuropeptide produced by endocrine cells in the digestive tract and that shares with CST the ability to interact with the growth hormone secretagogue receptor GHSR-1a[28]. Finally, a compound with a deletion of L-Lys14 (analogue **6**) was synthesised with the aim to improve stability, as mass spectrometry revealed this deletion as the main degradation product of the native neuropeptide. Unfortunately, the removal of Lys14 in analogue **6** reduced its aqueous solubility, thereby limiting its application in functional and structural assays.

Like the wild-type CST peptide, all analogues were prepared by standard Fmoc/ᵗBu solid-phase peptide synthesis (SPPS; Supplementary Fig. 1a) using commercial amino acids and Fmoc-protected Msa[29]. For the NMR structural studies, the peptides were used as TFA salt (0.5 mM) and for the in vitro and in vivo studies the TFA counter ion was replaced by acetate salts using ion exchange chromatography.

**Half-life in serum and in vitro activity of the CST analogues.** The stability of the natural CST-14 neuropeptide and of the analogues was evaluated by measuring their half-life in serum and in vitro affinity by analysing their binding preferences to Somatostatin receptors.

Since CST binds in vitro to the five known Somatostatin receptors[19,20,22,30,31] with nanomolar affinity, we examined whether these analogues maintained similar profiles. Analogues **2–3** had a high preference towards receptor 2, like all SST analogues currently on the market. Remarkably, analogues **4** and **5** showed distinct binding preferences for SST receptors in $IC_{50}$ assays with respect to SST, with analogue **5** exhibiting a binding preference for receptors 3, 4 and 5, and much lower binding to receptors 1 and 2 (Fig. 1d and Supplementary Fig. 1b). We also observed that analogue **5** showed higher stability in serum than native CST ($t_{1/2} = 21$ min vs. $t_{1/2} = 2$ min, respectively).

These results indicated that substitutions introduced in analogue **5** resulted in a 10-fold increase in stability in solution with respect to the native neuropeptide, thereby facilitating its storage and use. In contrast to the native CST, which preferentially selects receptor 2, analogue **5** showed a preference for SST receptors 3, 4 and 5.

**In vitro immunosuppressive and anti-inflammatory activity of the analogues.** We then studied the potential immunosuppressive

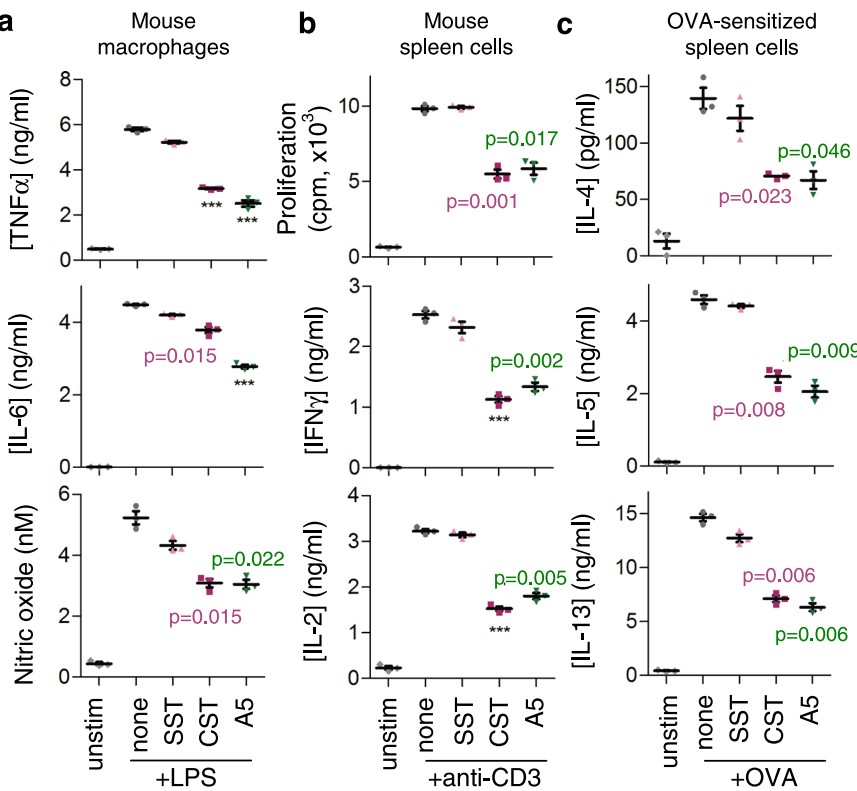

**Fig. 2 Cortistatin analogue 5 regulates macrophage and lymphocyte activation.** Effects of Somatostatin (SST), Cortistatin (CST) and analogue **5** (A5) on the production of inflammatory cytokines and nitric oxide by LPS-activated mouse Raw 264 macrophages (**a**), on cell proliferation and production of Th1-type cytokines by anti-CD3-activated mouse spleen cells (**b**) and on the production of Th2-type cytokines by OVA-activated spleen cells isolated from mice immunised with OVA (**c**). Peptides were used at 100 nM and the dose-response curve for analogue **5** and cortistatin are shown in Supplementary Fig. 2a, b. Cells cultured in medium alone were used as unstimulated controls. Data are the mean ± SEM with dots representing individual values of biologically independent cell cultures, each performed in duplicate. Statistical significance between groups was assessed by paired, two-tailed Student's t test. ***$p < 0.001$, ****$p < 0.0001$ vs. stimulated cells in the absence of peptides. Exact $p$-values are shown for $p > 0.001$. Source data are provided as a Source Data file.

and anti-inflammatory effects of our analogues, with CST and SST as controls, using various established in vitro assays[15,32–34]. CST and its analogue **5** markedly reduced the production of the inflammatory mediators TNFα, IL6 and nitric oxide (NO) by murine macrophages activated with a bacterial endotoxin (lipopolysaccharide (LPS); Fig. 2a), downregulated the proliferative response and the production of the Th1 cytokines IFNγ and IL2 by activated mouse spleen cells (Fig. 2b), and impaired the production of Th2 cytokines (IL4, IL5 and IL13) by ovalbumin-sensitised lymphocytes (Fig. 2c). We also compared the dose-response of CST and analogue 5, which showed similar effects at the same doses in activated mouse macrophages and spleen cells (Supplementary Fig. 2a, b). All these regulatory activities were not due to an effect of analogue 5 on the viability or apoptosis of activated lymphocytes and macrophages (Supplementary Table 2). In contrast, analogues 2, 3 and 4 showed no significant effects on the inflammatory responses in activated lymphocytes and macrophages (Supplementary Fig. 2c, d).

Our results indicate that analogue 5 exerted the strongest effect in the anti-inflammatory and immunosuppressive assays among the analogues tested.

**Effects in animal models of inflammatory bowel disease.** Having confirmed the immunosuppressive activity of analogue 5 in vitro, we next examined its potential therapeutic action in two established experimental models of acute and chronic colitis induced by oral administration of dextran sulfate sodium (DSS) and by intrarectal infusion of 2,4,6-trinitrobenzene sulfonic acid

(TNBS). These models display clinical, histopathological and immunological features of human UC and CD, respectively[15,34]

**DSS model.** In the DSS model, intestinal inflammation results from the impairment of the intestinal epithelial cell barrier function by DSS, subsequent exposure of the submucosa to various luminal antigens (bacteria and food) and activation of the inflammatory cells involved in innate immunity. By 7 days, oral administration of 5% DSS resulted in a progressive increase in the disease activity index, characterised by acute severe colitis, bloody diarrhea and sustained weight loss, resulting in 50% mortality (Fig. 3a). Colitis was correlated with dramatic signs of colon damage and inflammation, characterised by colon shortening and increased organ weight with respect to the control animals (Fig. 3a). Systemic treatment with analogue 5 for 3 days avoided mortality, ameliorated body weight loss, and improved the wasting disease and colon inflammation, similarly to the effects induced by native CST. Histological examination showed that treatment with analogue 5 and CST reduced the DSS-induced colonic transmural inflammation, mucin-producing goblet cell depletion, epithelial ulceration, focal loss of crypts and infiltration of inflammatory cells in the lamina propria (Fig. 3a).

Treatment with analogue 5 was also effective in a remitting-recurrent model of UC induced by administration of 3% DSS in two cycles. Subcutaneous injection of analogue 5 during the two acute peaks of colitis significantly reduced clinical activity (Fig. 3b). This effect was reflected by improvement of stool consistency, less rectal bleeding, decrease in colon shortening,

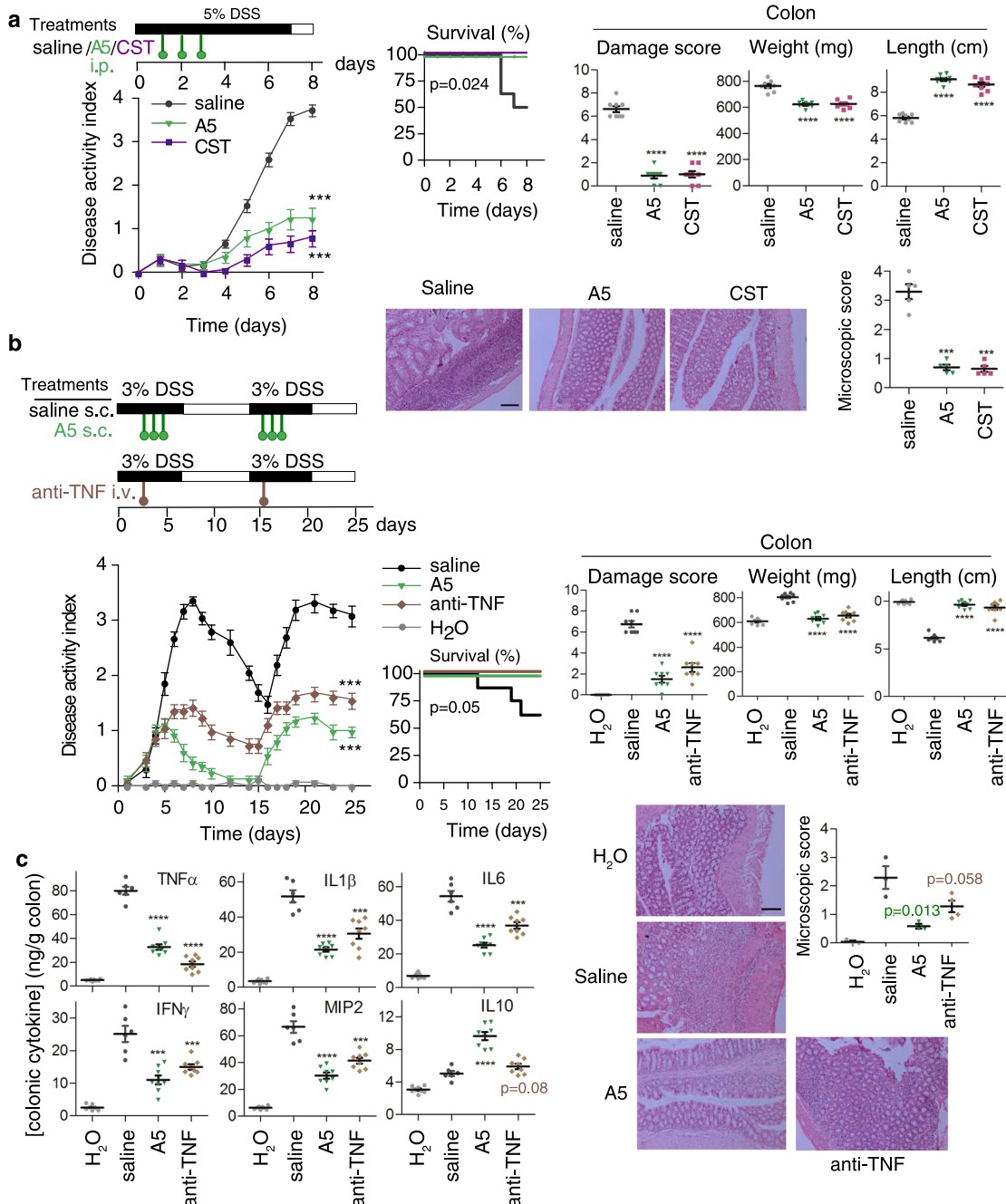

**Fig. 3 Therapeutic effect of analogue 5 on DSS-induced ulcerative colitis. a** Protective effects against acute severe colitis induced by oral DSS by the treatment with Cortistatin (CST) and analogue **5** (A5) (as indicated in the scheme) evaluated by disease activity indexes (scoring body weight loss, stool consistency and presence of faecal blood), survival rate, macroscopic signs of colon inflammation (damage score, length and weight of colon) and histopathological scores. **b** Comparative therapeutic effects of treatments with A5 and anti-mouse TNFα antibody in a model of relapsing-remitting colitis induced by cyclic administration of DSS (see scheme for experimental design). **c** Effects of A5 and anti-TNFα treatments on levels of cytokines in colonic mucosa of mice with DSS-induced relapsing-remitting colitis. Mice receiving tap water instead of DSS were used as naive controls. Animals injected with saline instead of A5 were used as untreated colitic mice. $n = 8$ mice/group for all assays, unless for histological analysis ($n = 5$ mice/group in panel **a**, and $n = 3$ mice/group in panel **b**, where representative images are shown at ×100 magnification, scale bar: 200 μm). Data are mean ± SEM with dots representing individual values of biologically independent animals. Statistical differences between groups were calculated using two-tailed non-parametric Mann–Whitney test (for disease activity index, colon damage and microscopic scores), unpaired two-tailed Student's *t* test (for colon length and weight) and Kaplan–Meier test (for survival). ***$p < 0.001$; ****$p < 0.0001$ versus untreated DSS-colitic mice (saline). Exact *p*-values are shown for $p > 0.001$. Source data are provided as a Source Data file. i.p. intraperitoneal, s.c. subcutaneous, i.v. intravenous.

amelioration of colon damage and histopathological signs, and by improvement in survival rate. Treatment with analogue **5** was even more efficient than a therapy of reference based on a neutralising anti-TNFα antibody (Fig. 3b). Interestingly, an initial injection of analogue **5**, but not of anti-TNFα antibody, only during the first peak of colitis conferred significant resistance to disease recurrence during a second cycle of DSS administration (Supplementary Fig. 3).

We next studied the in vivo effect of analogue **5** on the production of inflammatory mediators that are mechanistically linked to colitis. In accordance with clinical and histopathological signs, the colons of analogue **5**-treated mice with DSS-induced colitis showed reduced levels of inflammatory cytokines (TNFα, IFNγ, IL6, IL1β and IL17) and the chemokine macrophage inflammatory protein-2 (MIP-2), and increased levels of the anti-inflammatory/regulatory cytokine IL10, in comparison with untreated animals (Fig. 3c). As expected, treatment with the anti-TNFα antibody also reduced the mucosal levels of inflammatory mediators in colitic mice but failed to significantly increase the amounts of IL10 (Fig. 3c).

**TNBS model**. We also explored the protective and curative effects of analogue **5** in the progression of TNBS-induced acute and chronic colitis and compared the efficiency of different administration strategies. In this CD model, intestinal inflammation results from the initial rupture of the intestinal barrier using 50% ethanol. After this step, the haptenization of autologous host mucosal proteins is mediated by the action of TNBS and subsequent stimulation of a Th1 cell-mediated immune response against TNBS-modified self-antigens. The systemic injection of analogue **5** during the progression of the acute colitis significantly protected the animals from the profound and continuous body weight loss, bloody diarrhea and mortality caused by severe colonic inflammation (Fig. 4 and Supplementary Fig. 4). We observed that both intraperitoneal (i.p.) and subcutaneous (s.c.) administration routes showed similar effectiveness whereas oral (p.o.) administration was less effective at ameliorating TNBS-induced acute colitis (Supplementary Fig. 4). Interestingly, from a therapeutic perspective, although treatment with analogue **5** during the effector phase of the disease was highly protective (Supplementary Fig. 4a), its administration following a curative regimen at the peak of the disease also resulted significantly effective at reducing clinical and histopathological symptoms and improving survival (Supplementary Fig. 4b), thus showing comparable effects to those of treatments of reference, such as a neutralising TNF therapy or mesalazine (Fig. 4). Analysis of the production of inflammatory mediators yielded similar results to those obtained on the DSS model, with a broad anti-inflammatory activity of analogue **5** in the colon (Fig. 4c) accompanied by down-regulation of the systemic inflammatory response (Supplementary Table 3).

In a model of chronic colitis induced by weekly infusion of increasing doses of TNBS, repetitive treatments with analogue **5** after each TNBS infusion almost completely reversed disease progression and colon inflammation, in a similar way to that resulting from chronic administration of mesalazine or repetitive injections of the anti-TNFα antibody (Fig. 5). As previously observed in DSS-induced colitis, an initial injection of analogue **5** during the first week conferred significant protection from subsequent exposure to TNBS (Supplementary Fig. 5), thereby pointing to the induction of tolerance to disease recurrence. However, similar treatments with either mesalazine or the anti-TNFα antibody fully or partially failed to induce this tolerance effect (Supplementary Fig. 5).

**Effects on peripheral lymphoid organs**. Finally, we addressed whether analogue **5** ameliorates bowel inflammation by reducing autoreactive T-cell responses in the peripheral lymphoid organs that drain colonic mucosa. Mesenteric lymph nodes (MLNs) obtained from TNBS-treated mice showed marked proliferation and effector T cells producing high levels of Th1-type cytokines (IFNγ), especially upon restimulation (Supplementary Fig. 6b). In contrast, MLNs from analogue **5**-treated mice proliferated significantly less and produced low levels of IFNγ and high levels of the regulatory cytokine IL10 (Supplementary Fig. 6b). Similarly, determination of intracellular cytokine expression in MLN CD4 T cells of DSS-treated mice showed that analogue **5** decreased the number of IFNγ–producing Th1 cells and of IL17-producing Th17 cells, whereas it increased the number of IL10-secreting CD4 T cells (Fig. 6b). Moreover, treatment with analogue **5** decreased the systemic levels of anti-TNBS IgG (Supplementary Fig. 6c). This observation points to attenuation of antigen presentation and B-cell production of autoantibodies, a measure of the level of adaptive immunity. As expected from our in vitro experiments (Fig. 2), the addition of analogue **5** to MLN cultures from colitic mice suppressed T-cell proliferation and inhibited IFNγ production, while increasing the levels of IL10. This observation indicates that analogue **5** directly deactivated effector Th1 cells (Supplementary Fig. 6d). Given that several studies have demonstrated that IL10-secreting regulatory T cells (Treg) confer significant protection against IBD by decreasing the activation of autoreactive Th1 cells[35] and analogue **5** increased colonic IL10 levels (Figs. 3c, 4c and 5c) and MLN IL10-secreting CD4 cells (Fig. 6b), we examined the capacity of this analogue to increase the Treg population in colitic mice. Treatment with analogue **5** elevated the percentage of CD25$^+$FoxP3$^+$ Treg in MLN CD4 lymphocytes of mice with TNBS- and DSS-induced colitis (Fig. 6c and Supplementary Fig. 6d). In agreement with these findings, the administration of analogue **5** to mice with DSS-induced colitis decreased the gene expression of T-bet (a Th1 transcription factor) and RORγ-t (a Th17 transcription factor) and increased the expression of FoxP3 in colonic mucosa (Fig. 6d). These findings indicate that treatment with analogue **5** significantly inhibits the differentiation of autoreactive/inflammatory Th1 and Th17 cells and generates IL10-secreting T cells in this model.

Overall, on the basis of these results in animal models, analogue **5** emerges as a potential candidate for the treatment of IBD.

**Conformational studies of Cortistatin and its analogues by NMR**. To explain the different activity profiles of the analogues, we analysed their conformational properties by NMR, starting with analogue **5**, which was the most biologically active compound (Fig. 7a, b). The NMR data allowed us to characterise a major ensemble of conformations in solution due to well dispersed signals in the 2D TOCSY and to the presence of abundant and unambiguous NOEs in the 2D NOESY spectra. A dilution set of 1D experiments in 90%$H_2O$/10%$D_2O$ to illustrate the lack of aggregation at this range of concentrations is shown as Supplementary Fig. 7a, b. A set of NOEs indicates that Msa5, Phe6 and Phe10 aromatic rings are clustered on one side of the peptide plane, with the rings of Msa5 and Phe10 oriented edge-to-face, and with the Phe6 and Phe10 rings arranged offset-stacked, all pointing away from the D-Trp7:Lys8 pair (Fig. 7a, c, d). These NOEs were also present in the native CST neuropeptide but were more intense in analogue **5** (Figs. 1b and 7c), thereby suggesting that this analogue displays an enriched set of conformations already present in the pool of conformations sampled by the native neuropeptide.

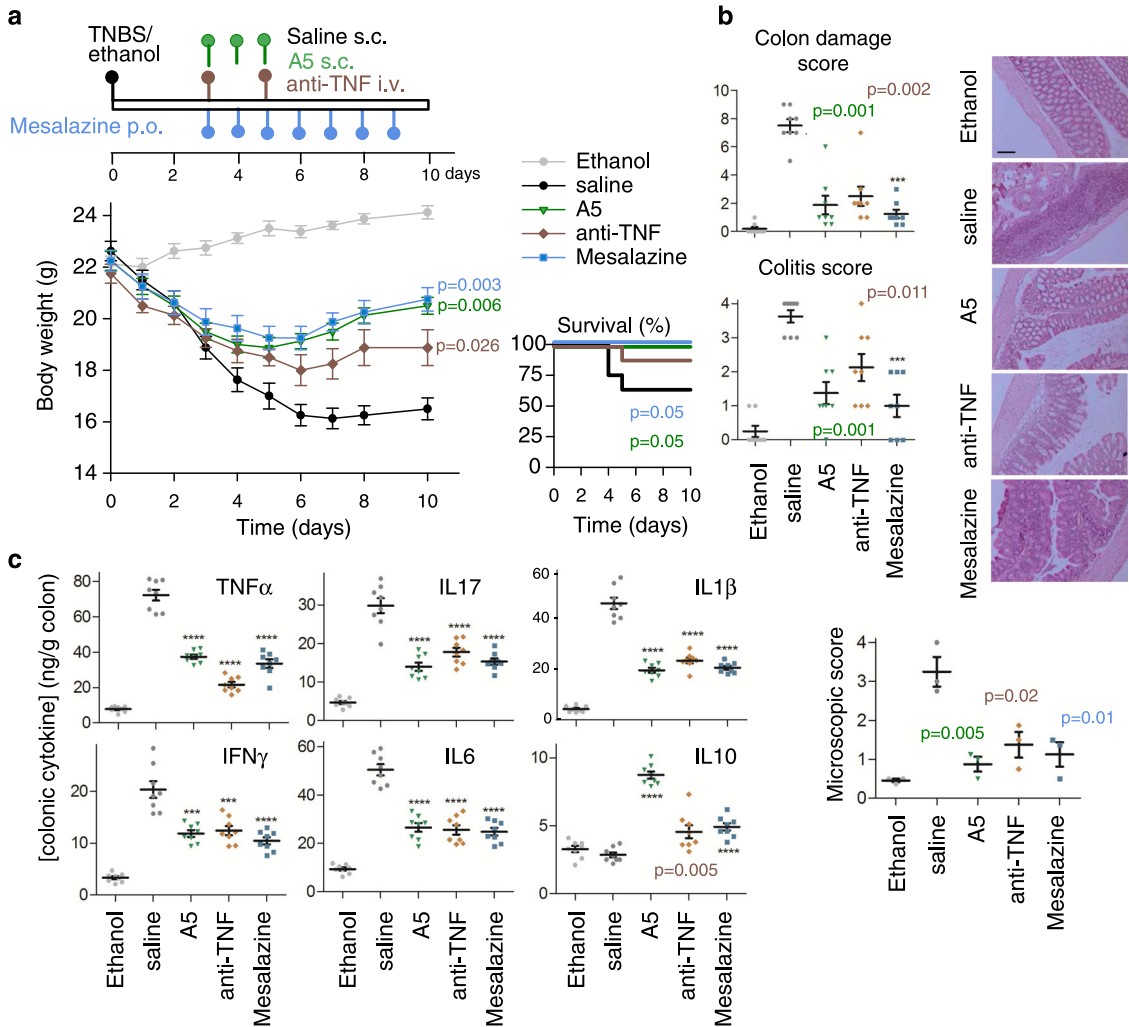

**Fig. 4 Comparative therapeutic effects of analogue 5 with treatments of reference on TNBS-induced acute colitis.** Acute colitis was induced in mice by intrarectal TNBS and the animals were then treated with analogue **5** (A5) or with treatments of reference (anti-TNFα antibody and Mesalazine) following the therapeutic regime indicated in the scheme (**a**). Mice injected intrarectally with 50% ethanol were used as basal controls. Animals injected with saline instead of A5 were used as untreated colitic mice. Disease evolution and severity were monitored by survival and weight loss, colitis score, macroscopic colon damage score and histopathological signs (**b**) and by measuring the cytokine levels in colonic mucosa (**c**). $n = 8$ mice/group in all assays, unless for histopathological analysis ($n = 3$ mice/group, where representative images are shown at ×100 magnification, scale bar: 200 μm). Data are mean ± SEM with dots representing individual values of biologically independent animals. Statistical differences between groups were calculated using two-tailed non-parametric Mann–Whitney test (for colon damage, colitis and microscopic scores), unpaired two-tailed Student's $t$ test (for body weight and colonic cytokines) and Kaplan–Meier test (for survival). ***$p < 0.001$; ****$p < 0.0001$ versus untreated TNBS-colitic mice (saline). Exact $p$-values are shown for $p > 0.001$. Source data are provided as a Source Data file. s.c. subcutaneous, i.v. intravenous, p.o. oral.

The backbone superposition of the 15 best conformers (selected according to structure validation tests and energy values) is shown as Fig. 7b whereas the side chain orientation of some selected amino acids and all side chains are shown in Fig. 7c, d. The orientation of the octanoyl moiety is not restrained in the calculation since only a few weak NOEs were observed from the octanoyl residue to Lys3/14 side chains. Remarkably, it seems that both Lys side chains shield the octanoyl moiety from the rest of the peptide surface (Fig. 7c), defining a hydrophobic pole adjacent to the aromatic cluster located in the middle of the structure. These hydrophobic areas do not prevent the peptide from being soluble but might enhance the interaction with CST natural binders/receptors in a cellular context, explaining why analogue **5** is more active than analogue **4**, which is identical in sequence but lacking the octanoyl moiety.

To analyse the structural characteristics of analogue **5**, we compared its backbone and aromatic cluster to those of other

analogues (and pharmacophores) described in the literature as SSTR binders[36–38]. To facilitate the comparison, we represented the Phi, Psi and Chi1 values for conserved residues (Supplementary Fig. 7c). These comparisons reveal that most analogues are very similar with respect to the backbone conformations in solution, specifically for residues surrounding the Trp-Lys area. To further compare the aromatic clusters and to define the potential pharmacophore in CST and in its analogue **5**, we also analysed the distances within the three aromatic residues 5, 6, 10 and also from these three residues to the conserved Trp7 and Lys8 pair (Fig. 7e and Supplementary Fig. 7d). For the measurements, we used the ensemble of the 15 best conformers and the C-γ position as the reference, because this position has been defined in the literature as the reference to describe pharmacophores of SST analogues[37,38]. This analysis revealed that analogue **5** and CST molecules share the presence of the aromatic cluster defined by residues 5, 6, and 10, although this

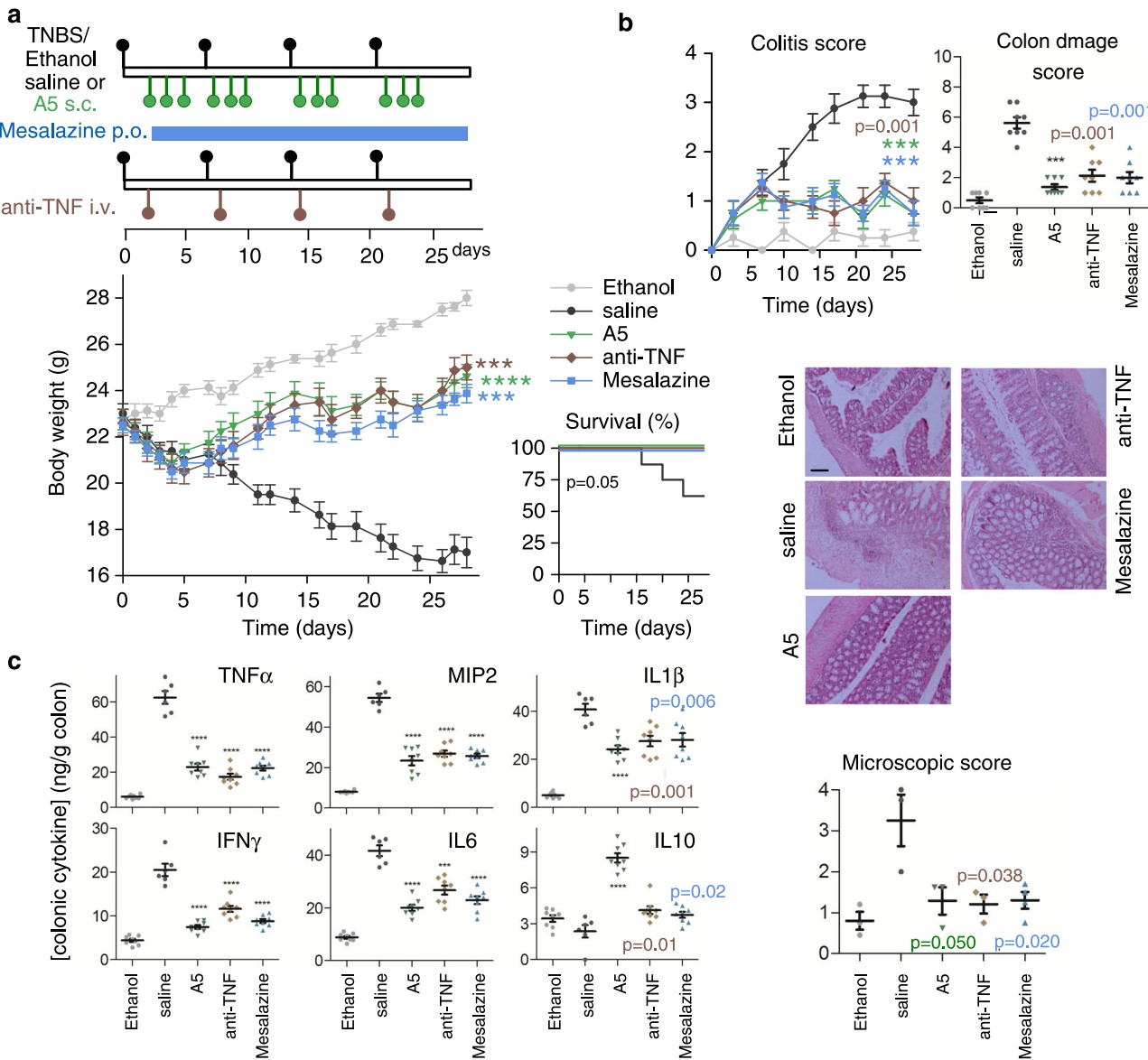

**Fig. 5 Therapeutic effects of analogue 5 on TNBS-induced chronic colitis.** Chronic colitis was induced in mice by intrarectal injections of increasing doses of TNBS once a week and animals were then repetitively treated with analogue **5** (A5), anti-mouse TNFα antibody or mesalazine as indicated in the scheme (**a**). Mice injected with 50% ethanol were used as basal controls. Animals injected with saline instead of A5 were used as untreated colitic mice. Disease evolution and severity were monitored by survival and weight loss, colitis score, macroscopic colon damage score and histopathological signs (**b**) and by measuring the cytokine concentration levels in colonic mucosa (**c**). $n = 8$ mice/group in all assays, unless for histopathological analysis ($n = 3$-4 mice/group, where representative images are shown at ×100 magnification, scale bar: 200 μm). Data are mean ± SEM with dots representing individual values of biologically independent animals. Statistical differences between groups were calculated using two-tailed non-parametric Mann–Whitney test (for colon damage, colitis and microscopic scores), unpaired two-tailed Student's $t$ test (for body weight, colon length and weight, and colonic cytokines) and Kaplan–Meier test (for survival). ***$p < 0.001$; ****$p < 0.0001$ versus untreated TNBS-colitic mice (saline). Exact $p$-values are shown for $p > 0.001$. Source data are provided as a Source Data file. s.c. subcutaneous, i.v. intravenous, p.o. oral.

cluster is highly enriched in analogue **5** with respect to that of native CST. This cluster seems to be unique with respect to the SST analogues previously synthesised[36–38]. The pharmacophore we propose recapitulates some features previously described (maintains the packing of the W-K pair) but includes specific features (shorter distances between the three aromatic residues of the cluster). It seems that this specific aromatic arrangement favours the recognition of SSTR3-5 but not that of SSTR2 receptor.

We also characterised the conformational properties of analogues **2**, **3** and **4**. NMR data acquired for them revealed a pattern of NOEs similar for these three analogues but different from analogue **5**. In these cases (Fig. 8a–c), the aromatic residues 5 and 10 align on one side of the molecule, whereas the side chain of residue 6 is located on the other side of the peptide plane. In this conformation, D-Trp7 is close to the pair of aromatic rings 5 and 10, due to the presence of weak NOEs involving the ring protons of D-Trp7 and the methyl protons of Msa5. Among the three compounds, analogue **3**, which also contains an octanoyl moiety and the Msa at position 6, is less rigid than analogues **2** and **4** (Fig. 8d–f). Moreover, in analogue **3** the number of NOEs between the pair D-Trp7:Lys8 is reduced, corroborating the presence of conformational variability in this part of the structure. Finally, we found that the loss of Lys14 (analogue **6**) had a

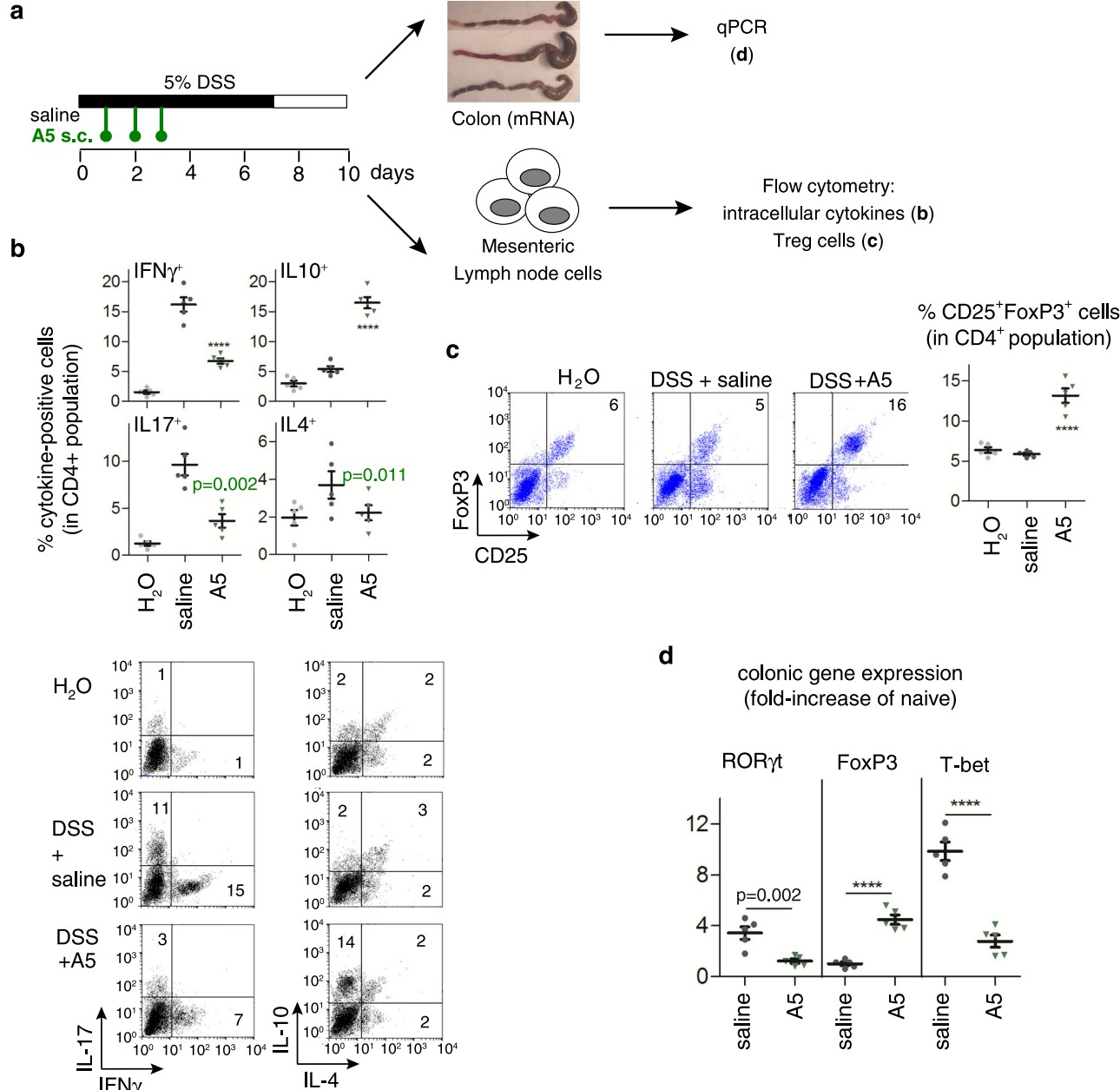

**Fig. 6 Treatment with analogue 5 reduces Th1 and Th17 cells and induces regulatory T cells in mice with ulcerative colitis.** Mice with DSS-induced acute colitis were treated with saline (untreated) or analogue **5** (A5) as indicated in the scheme (**a**). The percentages of cytokine-positive cells (**b**) and CD25⁺FoxP3⁺ Treg cells (**c**) in gated CD4⁺ population in mesenteric lymph node cells were analysed by flow cytometry (numbers in dot-plots represents percentages in each quadrant), and gene expression of T-bet, RORγt and FoxP3 in colon was determined by real-time qPCR (**d**). Mice receiving tap water instead of DSS were used as naive controls. $n = 5$ mice/group. Data are mean ± SEM with dots representing individual values of biologically independent animals. Statistical differences between groups were calculated using unpaired two-tailed Student's $t$ test. ****$p < 0.0001$ versus untreated DSS-induced colitic mice (saline). Exact $p$-values are shown for $p > 0.001$. Source data are provided as a Source Data file. FACS sequential gating/sorting strategies used for these experiments are shown as Supplementary Fig. 6f, g.

significant impact on the solubility of the sample, preventing the acquisition of 2D NMR experiments in aqueous buffer solution and explaining the poor activity of this analogue in the cellular assays.

All in all, our NMR studies indicate that the combination of Msa in position 5 and of the octanoyl moiety at the N-terminus contribute to favoring the cluster formation observed in analogue **5**, since the other analogues, either lacking such octanoyl moiety (analogue **4**) or with the Msa modification at a different position (analogue **2** and **3**), do not adopt this structural arrangement.

## Discussion

IBD is an emerging global and chronic condition, with patients being diagnosed prior to 35 years old and affecting children and adolescents worldwide. The treatments are complex because patients normally respond to different medications and very often to a combination of them. The rapid development of IBD runs along with demographic shifts (urbanisation) and an impact of environmental factors (diet, microbiota, pollution and psychological stress). Unfortunately, the exact causes of the disease are not fully understood, although the inflammation is often the result of

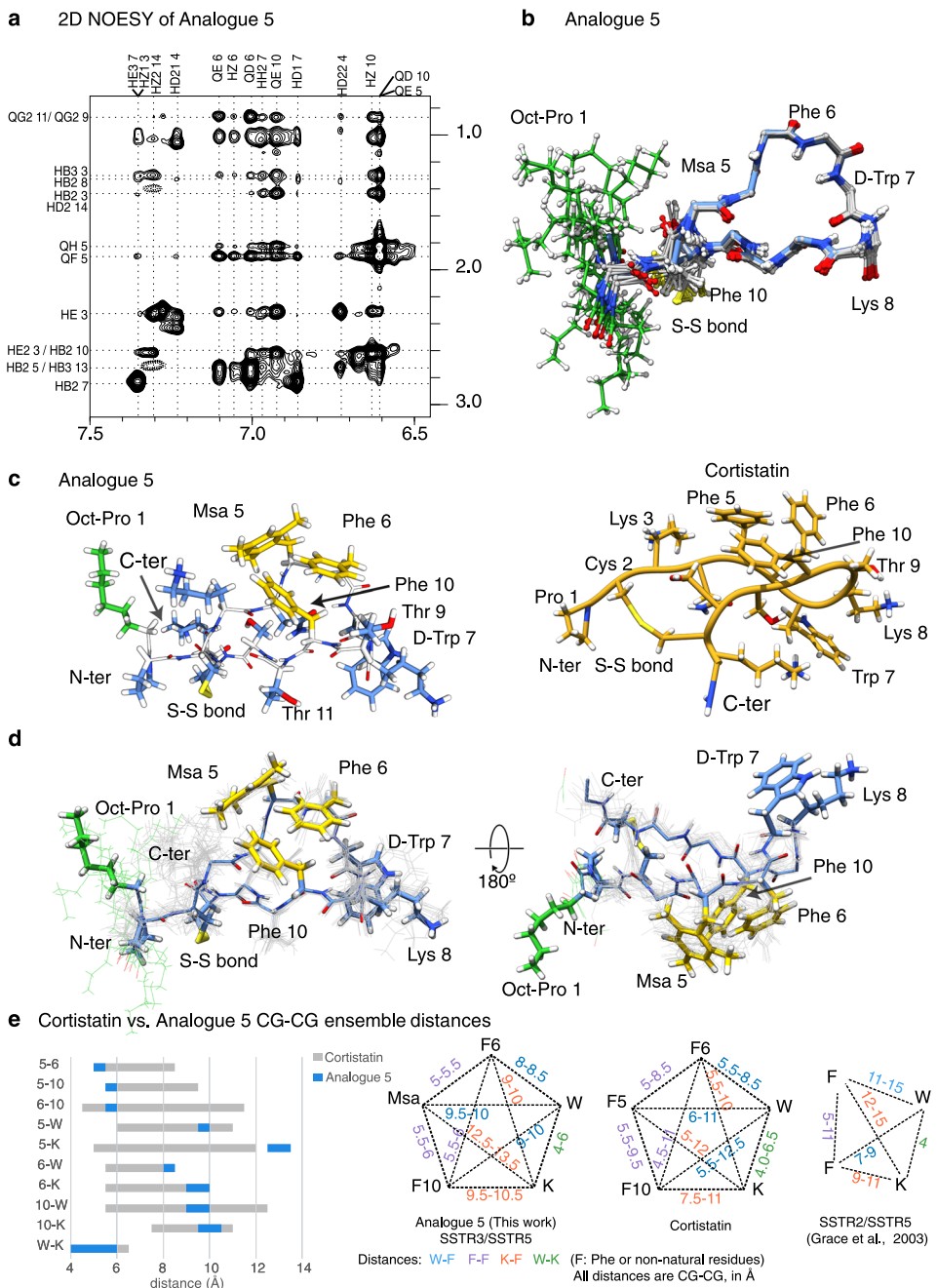

**Fig. 7 NMR conformers of analogue 5.** NMR restraints and ensemble of conformers corresponding to A**5**. **a** Side-chain amide region of the analogue **5** (¹H-NOESY experiment), with peak assignments. Amide/proton alpha (left) and amide/aromatic (right) regions of the analogue **5** (¹H-NOESY experiment), with peak assignments are shown as Supplementary Fig. 7a. 1D set of dilution experiments are included as Supplementary Fig. 7b. **b** Backbone superposition of the 15 lowest-energy calculated conformers. **c** Analogue **5**, lowest energy conformer. Side chains are depicted as sticks. The residues defining the aromatic cluster are shown in yellow, the octanoyl moiety is highlighted in green whereas the remaining residues are shown in blue. The peptide backbone is shown in white. On the right, the equivalent Cortistatin conformer for comparison (shown in Fig. 1b). **d** Side chain and backbone distribution for the ensemble of the 15 best conformers shown as two orientations. Some residues are labelled. **e** Comparison of the pharmacophore region of analogue **5** to that of Cortistatin conformations represented in Fig. 1b. The comparison includes the distances between C-gamma atoms of the five residues considered to define the pharmacophore. The backbone and Chi1 comparison of these six molecules are provided as Supplementary Fig. 7c. Distances are represented as ranges to include the distance dispersion in the ensemble of conformations.

a combination of genetic factors and other effects that alter the immune system.

To expand the repertoire of molecules to be used to treat patients, we have focused on the potential applications of the neuropeptide CST. CST is expressed in immune cells and it is a potent anti-inflammatory agent that can deactivate some IBD responses in animal models[10]. The native sequence has other

roles in modulating the GH-releasing activity of ghrelin and parallels some roles of SST. However, the pharmacological applications of CST (and other natural endogenous hormones) are often a challenge due to rapid proteolysis in plasma[39]. In the case of CST, its half-life is limited to minutes and although this time might not be an issue in assays with small animal models, it can compromise the efficient distribution and curative effects of

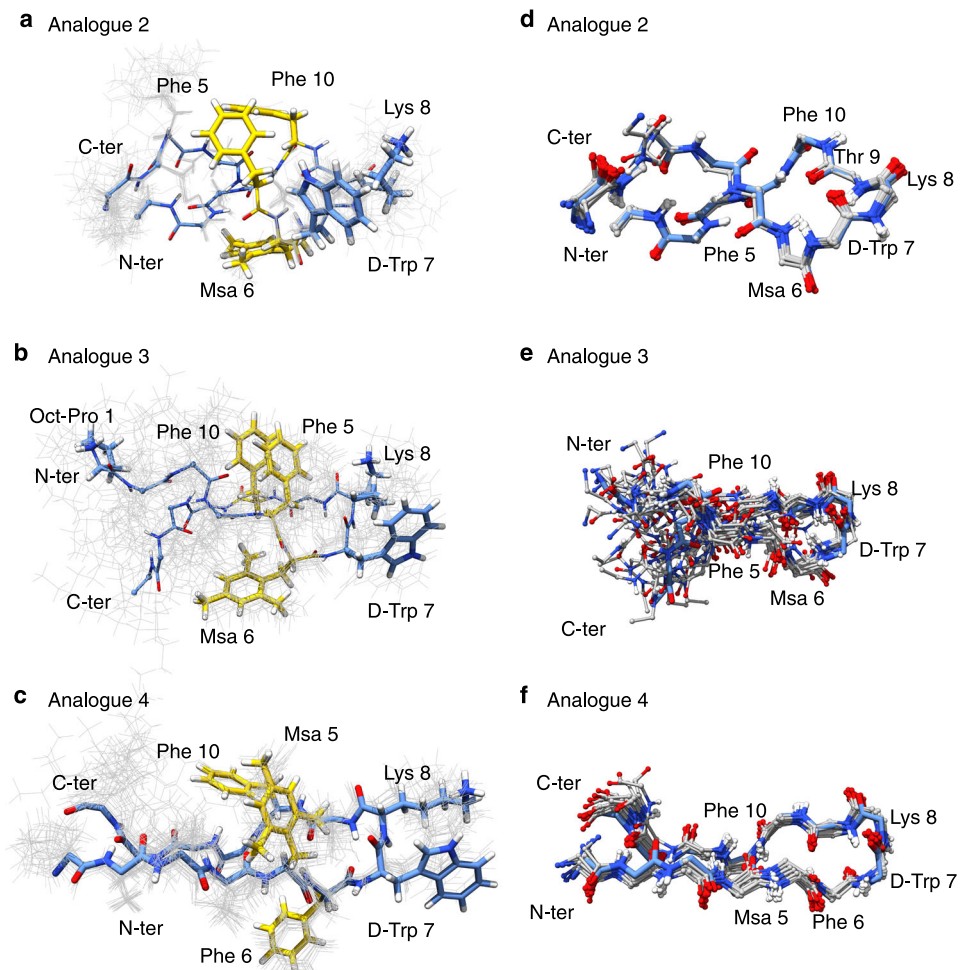

**Fig. 8 NMR conformers corresponding to analogues 2, 3 and 4. a** Analogue **2**, side chain and backbone distribution for the ensemble of the 15 best conformers. Side chains are depicted and labelled. The lowest energy conformer is highlighted as sticks. **b**, **c** as **a**, for analogues **3** and **4**. **d** Backbone superposition of the 15 best calculated conformers for analogue **2**. **e**, **f** as **d**, for analogues **3** and **4**.

this peptide when administered to larger species like humans. In addition, its low stability poses a challenge to its storage in solution, as well as to the administration mode itself since the solution has to be prepared immediately before injection in order to maintain its therapeutic effects.

To overcome these limitations, we examined the capacity of CST-based analogues as potential pharmacological agents that can retain the anti-inflammatory capacity of the native neuropeptide but show increased stability. We found that analogue **5** displayed a 10-fold stability increase in serum while preserving the pattern of immunosuppressive and anti-inflammatory activity of CST in vitro and in animal models. This analogue contains a Msa in position 5 (a modification of the phenylalanine ring), a D-Trp at position 7 and an octanoyl moiety at the N-terminus as the main modifications of the native sequence. Using NMR, we observed that this peptide displays an aromatic cluster in solution, defined by contacts between the aromatic rings of Phe6, Phe10 and Msa5, which are absent in the other analogues but present in the ensemble of conformations of native CST. We suggest that receptors or binding partners present in inflammatory cells might recognise epitopes similar to the aromatic cluster found in analogue **5**.

With respect to binding to SST receptors, CST is known to interact with all of them in the low nanomolar range[7]. The $IC_{50}$ values determined here reveal that analogue **5** shows preferential binding to receptors 3, 4 and 5 (instead of receptor 2, the

preferred target of SST analogues). We hypothesise that this effect could be due to its specific cluster of aromatic residues (Fig. 7e), opening the possibility of using this analogue to inhibit prolactin secretion in prolactin-secreting adenomas since these roles are ascribed to specific functions of the SST receptor 5[40,41].

Several potential mechanisms can explain the therapeutic effects of analogue **5** on the effector phase of colitis. To explore some of these mechanisms, we evaluated the immunosuppressive and anti-inflammatory activities of the analogues following the levels of cytokines and inflammatory factors in vitro. These results were corroborated by experiments performed in mouse models of IBD. In both IBD and experimental colitis, activated Th1 and Th17 cells promoted exacerbated macrophage and neutrophil infiltration and activation, which gave rise to prolonged and uncontrolled production of inflammatory cytokines and chemokines in intestinal mucosa and colonic tissue destruction. As previously described for native CST[9,10,42], analogue **5** may regulate the two arms of the immune response involved in the pathogenesis of IBD. In fact, this analogue strongly reduces colonic inflammation by downregulating the production of a plethora of inflammatory/cytotoxic mediators and Th1/Th17-type cytokines. Our in vitro experiments show that analogue **5** directly deactivates macrophages and T-cell responses. The inhibition of Th1 and Th17 responses might be the result of a direct action of the draining MLNs on cells, because the MLN cells obtained from analogue **5**-treated animals

were refractory to restimulation under Th1 conditions in vitro and showed reduced numbers of Th1 and Th17 populations. In contrast, treatment with analogue **5** increased the production of IL10 in MLN CD4 T cells and the levels of IL10 in colonic mucosa. IL10 is an anti-inflammatory cytokine used as a signature for a subset of Treg cells that exert suppressive functions on self-reactive T lymphocytes and induce peripheral tolerance in vivo[35]. Indeed, an initial short treatment of experimental IBD with analogue **5** increased the number of MLN Foxp3 + CD4 + CD25 + Treg cells and the gene expression of colonic FoxP3, as well as inducing tolerance to colitis recurrence. The regulation of a wide panel of immune mediators by analogue **5** represents a therapeutic advantage. In fact, our results revealed that treatment with this molecule was more effective than the administration of the anti-TNFα antibody used as the reference treatment in these assays, thereby highlighting the potential of analogue **5**.

We believe that our work, based on a family of compounds derived from natural peptides, opens a path to the rational design of drugs to help control inflammatory bowel syndromes.

## Methods

**General syntheses**. The syntheses were performed by SPPS on 0.25 g of 2-Cl-Trt resin (1.60 mmol/g) using the Fmoc/tBu strategy. The C-terminal first amino acid (1.0 equiv) was coupled in the presence of DIPEA (3.0 equiv) in 1.25 mL of DCM as solvent for 45 min and finally end-capped with methanol (0.8 mL/g). Then, Fmoc removal was performed by treating the peptidyl resin with 20% piperidine in sDMF ($1 \times 1'$ and $1 \times 5'$). The next amino acids (2.5 equiv or 1.5 equiv for Fmoc-L-Msa-OH) were coupled using DIPCDI and HOBt (2.5 equiv respectively), as activating reagents, in DMF for 40–60 min. We have previously reported the synthesis of Fmoc-L-Msa-OH modified amino acid[29].

Kaiser Test was used to check coupling completion. This procedure was repeated for the following Fmoc-protected amino acids and for the last Boc-Pro-OH in compounds **1**, **2** and **4**, and Boc-L-Ala-OH in compound **7**. The cleavage of the fully protected linear peptide from the resin was carried out using a cleavage cocktail (DCM:TFE:AcOH, 70:20:10 (v/v/v)) for 2 h. The formation of the disulfide bridge in all the analogues was achieved using iodine (0.73 g, 10 equiv) in 3.57 mL of DCM:TFE:AcOH solution at room temperature for 15′ and then quenched with an aqueous solution of sodium thiosulphate (1.52 g, 22 equiv) in 6.12 mL of water. The aqueous layer was extracted with DCM ($3 \times 4$ mL), the combined organic layer was washed with a mixture of an aqueous citric acid 5% solution/sodium chloride (1:1) and evaporated under reduced pressure. Finally, total deprotection of the side-chains was performed using 6.8 mL of acidic mixture (TFA/DCM/Anisole/$H_2O$, 55:30:10:5 (v/v)) for 4 h. Then, the remaining solution was washed with heptane (13 mL) and the aqueous phase was precipitated in $Et_2O$ (−10 °C). The obtained suspension was filtered through a filter plate and the filtrates were discarded. The residue was washed with ether, discarding the filtrates in each washing.

The crude was purified by semipreparative system equipped with a NW50 column filled with silica Kromasil 10 microns. The peptide was suspended in 0.1 N AcOH and DOWEX resin fitted out in 0.1 N AcOH was added. The final acetate compound was recovered by filtration and characterised by mass spectrometry in ESI-MS equipment and NMR.

**NMR methods**. NMR experiments were recorded at 285 K using a Bruker Avance III 600-MHz spectrometer equipped with a quadruple resonance z-gradient CryoProbe (5 mm QPQCI). The data was processed with TopSpin version 3.5 and chemical shift assignments were generated using Cara[43] and a combination of 2D TOCSY and NOESY homonuclear experiments. Distance restraints derived from the NOESY experiments were used for structure calculation using unambiguously assigned peaks. The structures were calculated with the programme CNS 1.2[25]. The protocol[44] consisted of an implicit water simulated-annealing of 120 structures using 8000 cooling steps followed by an explicit water refinement of the calculated structures using all experimental restraints during 1200 steps.

**Binding assays**. All peptides were subjected to anion exchange ($F_3CCOO^-$ to $AcO^-$) by using DOWEX resin (DowexMonosphere 550 A (OH)). Biological tests were carried out with purified peptides containing acetate counter-ion.

Human recombinant CHO-K1 cells expressing independently each of the five somatostatin receptors (SSTR1-SSTR5) were used. Cells were incubated in HEPES buffer pH 7.4 with the analogues and controls at 10 different concentrations in a range of 1 pM to 10 μM for 2–4 h, in order to rank the analogues' preferences. $^{125}$I-Tyr11 SST14 was used as radio-ligand and SST as cold ligand. Membranes were filtered and washed three times and the filters were counted to determine [$^{125}$I]-SST-14 specifically bound (performed at Eurofins Panlabs, Inc.). The radioactivity measured in the absence of SST is considered as total binding and the effect measured in the presence of 1 μM of SST is considered as nonspecific, being the

specific binding the difference between total and nonspecific binding. Results are presented as the inhibition of specific binding (percentage inhibition) as a mean of duplicates with error bars representing the 95% confidence interval of the curve fitting. Graphical and tabular representation of the percentage inhibition of specific binding, as $IC_{50}$ (nM) binding values for the analogues against the five SSTR1-SSTR5 is represented in Supplementary Fig. 1b and Supplementary Table 1. The reliability of the assay was assessed by using SST reference standards that were run as an integral part of each binding assay with the five (SSTR1-SSTR5). $IC_{50}$ values were determined by a non-linear, least squares regression analysis using Prism 9 (GraphPad Software, USA), using the Dose-response inhibition (three-parameters) model.

**Determination of in vitro stability of peptides in human plasma**. The peptides were dissolved in water at a concentration of 6 mg/mL and warmed up to 37 °C. Human plasma was obtained as a lyophilised solid (K3 EDTA Plasma, BBI solutions, code S112-1), reconstituted with sterile 0.9% sodium chloride solution and stored at −20 °C. Human plasma was thawed and incubated at 37 °C before use.

The peptides were incubated in 90% human plasma at 37 °C for different times and then precipitated with two volume equivalents of methanol. The samples were cooled in an acetone-carbon dioxide bath for a few seconds and centrifuged for 12 min at 4 °C at ~9000×g. The supernatant was filtered with a 0.45-μm PVDF filter and analysed in triplicate by RP-HPLC using an isocratic method (eluent A = 0.1% TFA in water; eluent B = 0.07%TFA in ACN, column = Kromasil C8, 100 Å, 5 μm, $250 \times 4.6$ mm, flow = 1 mL/min, wavelength = 220 nm, injection volume = 20 μL, $T^a = 60$ °C). The disappearance of the peptide was determined in relation to the area of the initial time for calculating its half-life.

**Inflammatory response in vitro**. Inflammatory response was assayed in vitro on the mouse macrophagic cell line RAW 264 macrophages and mouse primary peritoneal macrophages. RAW 264 cells (ATTC, TIB-71) were grown at $5 \times 10^5$ cells/mL in complete DMEM medium (DMEM supplemented with 100 U/mL penicillin/streptomycin, 2 mM L-glutamine, 50 μM 2-mercaptoethanol and 10% heat-inactivated fetal calf serum) until reaching a confluence of 80% and then incubated in the absence (used as baseline reference) or presence of LPS (1 μg/mL, from E. coli serotype 055: B5, Sigma). Cortistatin (mouse/rat CST-14, Bachem) and its synthesised analogues were added at a concentration of 100 nM at the beginning of culture. After 24 h of culture, supernatants were harvested. Levels of the inflammatory cytokines TNFα and IL6 were determined by a specific sandwich ELISA by using capture/biotinylated detection monoclonal antibodies from BD Pharmingen (for TNFα: capture antibody clone G281-2626, cat # 551225; biotin antibody clone MP6-XT3, cat # 554415; for IL6: capture antibody clone MP5-20F3, cat # 554398; biotin MP5-32C1, cat # 554402) according to the manufacturer's recommendations. Nitric oxide content in culture supernatants was determined by quantifying nitrite levels using the Griess assay.

Resident primary macrophages were obtained from C57Bl/6J mice (male, 8-weeks-old, Charles River) by peritoneal lavage with DMEM medium. Peritoneal cells were washed in cold DMEM medium and incubated in complete DMEM medium at a concentration of $10^6$ cells/mL in 24-well plates ($2 \times 10^6$ cells/well). After 2 h at 37 °C, nonadherent cells were removed by extensive washing. At least 95% of the adherent cells were macrophages (~$10^6$ cells/well) as judged by morphological and phagocytic criteria and by flow cytometry. Macrophage monolayers were incubated with complete DMEM medium in the absence (unstimulated) or presence of LPS (1 μg/mL, from E. coli serotype 055:B5). Peptides (between 1 nM and 500 nM) were added at the initiation of the culture, cell-free supernatants were collected after 24 h of culture and cytokine and nitric oxide levels were determined as described above.

**Immune responses in vitro**. Spleen cells were isolated from C57Bl/6J mice (male, 8-weeks-old, Charles River) by mechanical cell dissociation, filtration using a nylon filter and red cell lysing. The spleen cells were incubated in complete DMEM medium at a density of $10^6$ cells/mL for 2 h. Nonadherent cells (consisting in 80% of T cells) were cultured in complete DMEM medium and stimulated with anti-mouse CD3e antibody (2 μg/mL, clone 145-2C11, BD Pharmingen, cat #: 553058) in the presence of different Cortistatin analogues (100 nM). After 48 h, culture supernatants were collected and the levels of cytokines (IFNγ and IL2) were determined by a specific sandwich ELISA by using capture/biotinylated detection monoclonal antibodies from BD Pharmingen (for IFNγ: capture antibody clone XMG1.2, cat # 554408; biotin antibody clone R4-6A2, cat # 551506; for IL2: capture antibody clone S4B6, cat # 554375; biotin antibody clone JES6-5H4, cat # 554426) according to the manufacturer's recommendations. To determine the effect of different analogues of Cortistatin in proliferation, the cells were cultured for 72 h, 0.5 μCi (0.0185 MBq)/well of [$^3$H]-thymidine was added for the last 8 h of culture, harvested the membranes and [$^3$H]-thymidine incorporated was quantified by scintillation counting.

To induce Th2 responses, we injected intraperitoneally Balb/c mice (male, 7-weeks-old, Charles River) with 200 μL of an emulsion 1:1 of ovalbumin (chicken OVA from Sigma-Aldrich; 100 μg OVA/mouse) and Imject Alum (ThermoFisher Scientific, an aqueous solution of aluminium hydroxide and magnesium hydroxide). Fourteen days after OVA/Alum injection, spleen cells were obtained by

mechanical cell dissociation, filtration through a nylon mesh and lysis of red blood cells. Spleen cells were incubated with complete DMEM medium at a density of $10^6$ cells/mL in 24-well plates. After 2 h at 37 °C, nonadherent cells (consisting in 80% of T cells) were replated in 24-well flat-bottom plates at $10^6$ cells/well in complete DMEM medium and stimulated with freshly dissolved OVA/PBS (100 μg/mL) in the absence or presence of analogue 5 or Cortistatin (at 100 nM). After 96 h, cell-free supernatants were collected and the levels of cytokines were determined by a specific sandwich ELISA by using capture/biotinylated detection monoclonal antibodies from BD Pharmingen (for IL4: capture antibody clone 11B11, cat # 554432; biotin antibody clone BVD6-24G2, cat # 554398; for IL17: capture antibody clone TC11-18H10, cat # 560268; biotin antibody clone TC11-8H4, cat # 555067) or by using R&D Systems Quantikine ELISA kits (for IL5 determination: cat # M5000; and for IL-13 determination; cat # M13000CB) according to the manufacturer's recommendations.

**Induction of acute and chronic ulcerative colitis with dextran sulfate sodium.** Acute colitis was induced in male C57Bl/6J mice (7-weeks-old, Charles River) by administering 5% DSS (molecular weight 20,000 Da; Sigma) from day 0 to day 7 in the drinking water ad libitum. At days 1, 2 and 3, Analogue 5 (at 5 nmol/mouse, 0.4 mg/kg, in 100 μL saline) was injected intraperitoneally (i.p.) in the DSS-treated animals. Cortistatin (mouse/rat CST-29, 5 nmol/mouse, from Bachem) injected i.p. at days 1, 2 and 3 was used as a product of reference. Recurrent-remitting colitis was induced in male C57Bl/6 mice (7-weeks-old, Charles River) by administering 3% DSS (molecular weight 20,000 Da; in the drinking water ad libitum) in two cycles of 7 days (from day 0 to day 7 and from day 15 to day 21 with normal water between cycles). Animals hydrated with normal water were used as naive controls. Analogue 5 (0.4 mg/kg, in 100 μL saline) was injected subcutaneously (s.c.) in the DSS-treated animals following two profiles: during the first cycle of DSS at days 3, 4 and 5; or during the two cycles of DSS at days 3, 4, 5, 16, 17 and 18. We used as a treatment of reference, the intravenous (i.v., in 20 μL saline through the tail vein) injection of a neutralising anti-mouse TNFα antibody (5 mg/kg, clone TN3-19.12 from BD Bioscience, cat #: 557516) at days 3, or at days 3 and 16.

In both acute and chronic models, colitis severity was assessed daily by scoring (scale 0–4) the clinical disease activity by evaluating stool consistency, presence of faecal blood and weight loss, including a summation of the three components: weight loss (0 = 0%, 0.5 = 1–10%, 1 = 11–15%, 1.5 = 16–20%, 2 ≥ 20%), diarrhea (0 = normal stool, 0.5 = soft stool and minimal wet anal fur/tail, 1 = diarrhea and moderate-to-severe wet anal fur/tail) and frank rectal bleeding (0 = absent, 0.5 = present but minimal, 1 = moderate/severe). Mice were killed on day 8 (acute model) or on day 25 (chronic model), the entire colon was removed from the caecum to the anus, and colon length and weight were measured as indirect inflammation markers. The macroscopic colonic damage score (scale 0–8) was assessed based on the grade of tissue adhesion, presence of ulceration and wall thickness: ulceration (0 = normal appearance, 1 = focal hyperemia, no ulcers, 2 = ulceration without hyperemia or bowel wall thickening, 3 = ulceration with inflammation at 1 site, 4 = two or more sites of ulceration and inflammation, 5 = major sites of damage extending >1 cm along length of colon), adhesions (0 = no adhesions, 1 = minor adhesions, colon can be easily separated from the other tissues, 2 = major adhesions) and thickness (maximal bowel wall thickness, in mm, measured with a caliper). After macroscopic examination, various colonic segments were immediately frozen in liquid nitrogen for protein extraction and cytokine determination, for RNA extraction and qPCR analysis, or for histopathological analysis as described below. In some animals, mesenteric lymph nodes (MLNs) were isolated at day 10 after initiating DSS administration and assayed for proliferation, cytokine production and flow cytometric analysis as described below.

**Induction of acute and chronic colitis with trinitrobenzene sulfonic acid.** To induce acute colitis, 2,4,6-TNBS (3 mg; Sigma) in 50% ethanol (100 μL) was administered intrarectally in 6-8-weeks-old male Balb/c mice (150 mg of TNBS/kg mouse) under light halothane-induced anaesthesia. Control mice received 50% ethanol alone. Animals were treated i.p. or s.c. with vehicle (saline) or with ana-logue 5 (0.4 mg/kg, 9 μg/mouse, in a total volume of 100 μL) 12 h, 24 h and 36 h (in a protective regime) after instillation of TNBS or 3, 4 and 5 days after TNBS (in a curative regime). In addition, analogue 5 was administered via oral gavage in a curative regime (15 mg/kg) twice/day at days 3, 4 and 5. As treatments of reference, anti-TNFα antibody (5 mg/kg, clone TN3-19.12) was injected i.v. at days 3 and 5, and Mesalazine (5-aminosalicylic acid, 50 mg/kg, Sigma) was administered via oral gavage, twice/day from day 3 to day 10. To induce chronic colitis, Balb/c mice (male, 7-weeks-old, Charles River) were repetitively injected intrarectally with TNBS (0.8 mg in 100 μL 50% ethanol at day 0; 1 mg in 100 μL 50% ethanol at day 7; 1.2 mg in 100 μL 50% ethanol at day 14 and 1.5 mg in 100 μL 50% ethanol at day 21). Controls were injected intrarectally with 50% ethanol (100 μL) at days 0, 7, 14 and 21. Analogue 5 was injected s.c. (0.4 mg/kg, in total volume of 100 μL saline), following two regimens of administration: tolerance treatment at days 3, 4 and 5, and chronic treatment at days 3, 4, 5, 8, 9, 10, 15, 16, 17, 22, 23 and 24. Anti-mouse TNFα antibody and Mesalazine were used as treatments of reference. Mesalazine was administered via oral gavage (50 mg/kg, twice a day, in 100 μL saline) in a tolerance treatment (from days 3 to 6) or in a chronic treatment (from days 3 to 28). Anti-TNFα antibody was injected i.v. (5 mg/kg) in a tolerance treatment (at days 3 and 5) or in a chronic treatment (at days 3, 8, 15 and 22).

In both TNBS-induced acute and chronic colitis, animals were daily monitored for the appearance of diarrhea, body weight loss and survival. At different time points, colitis signs were scored in base of stool consistency and rectal bleeding by two blinded observers using the following scale: 0 = normal stool appearance, 1 = slight decrease in stool consistency, 2 = moderate decrease in stool consistency, 3 = moderate decrease in stool consistency and presence of blood in stools, 4 = severe watery diarrhea and moderate/severe bleeding in stools. At day 10 (acute model), at day 28 (chronic model) or immediately after death of each animal, serum and colons were collected and evaluated for macroscopic damage (scale 0–10) based on criteria reflecting inflammation (hyperemia, bowel thickening and extent of ulceration) in a blinded fashion by two researchers: ulceration (0 = normal appearance, 1 = focal hyperemia, no ulcers, 2 = ulceration without hyperemia or bowel wall thickening, 3 = ulceration with inflammation at 1 site, 4 = two or more sites of ulceration and inflammation, 5 = major sites of damage extending >1 cm along length of colon and 6–10 = when an area of damage extended >2 cm along length of colon, score is increased by 1 for each additional cm of involvement). After macroscopic examination, various colonic segments were immediately frozen in liquid nitrogen for protein extraction and cytokine determination or for histopathological analysis as described below. In some animals, MLNs were isolated 10 days after TNBS injection and assayed for proliferation, cytokine production and flow cytometric analysis as described below. Moreover, sera were collected by cardiac puncture at day 10 after TNBS injection and the levels of cytokines and anti-TNBS IgG were determined by specific ELISAs as described below.

**Cytokine determination in serum and colon mucosa.** Protein extracts were isolated by homogenisation of colonic segments (50 mg tissue/mL) in 50 mM Tris-HCl, pH 7.4, with 0.5 mM DTT, and 10 μg/mL of a cocktail of proteinase inhibitors (all from Sigma-Aldrich). Samples were centrifuged at 30,000×$g$ for 20 min and stored at –80 °C until cytokine determination. Cytokine and chemokine levels in the serum and colonic protein extracts were determined by specific sandwich ELISAs using capture/biotinylated detection monoclonal antibodies from BD Pharmingen (for TNFα, IL6 and IFNγ as described above, and for IL10: capture antibody clone JES5-16E3, cat # 554463; biotin antibody clone SXC-1, cat # 554423) and polyclonal antibodies from Preprotech (for MIP-2: capture antibody cat # 500-P130; biotin antibody cat # 500-P130Bt; for IL1β: capture antibody cat # 500-P51; biotin antibody cat # 500-P51Bt) according to manufacturers' recommendations.

**Determination of anti-TNBS IgG levels in serum.** Anti-TNBS IgG were determined in sera by specific ELISAs as previously described[45]. Briefly, ovalbumin (100 mg) was dissolved in carbonate-bicarbonate buffer (50 mM, pH 9.6) and incubated with 1% TNBS for 2 h at 20 °C. 2,4,6-trinitrophenyl-ovalbumin was dialysed in PBS and then coated at 0.1 mg/mL in carbonate-bicarbonate buffer (50 mM, pH 9.6) in 96-well microtiter plates. After blocking with PBS/3% BSA, plates were incubated with diluted serum for 2 h at 20 °C. For detection, plates were incubated with horseradish peroxidase-conjugated secondary anti-mouse IgG policlonal antibodies (Jackson Immunoresearch, cat # 115-035-072) for 1 h at 20 °C and then TMB peroxidase substrate (ThermoFisher). Results were expressed in units relative to an internal control consisting in sera of mice that were immunised with 1% TNBS in Freund's complete adjuvant.

**Histopathological analysis.** A colon specimen from the middle part was fixed in 10% buffered formalin phosphate, embedded in sucrose, frozen in dry ice using optimal cutting temperature (OCT) compound and cryosectioned. Sections were stained with hematoxylin-eosin and examined in an Olympus microscope. Intestinal inflammation was graded from 0 to 4 as follows, in a blinded fashion by two independent researchers: 0, no signs of inflammation; 1, low leucocyte infiltration; 2, moderate leucocyte infiltration; 3, high leucocyte infiltration, moderate fibrosis, high vascular density, thickening of the colon wall, moderate goblet cell loss and focal loss of crypts; and 4, transmural infiltrations, massive loss of goblet cell, extensive fibrosis and diffuse loss of crypts.

**MLN cell cultures.** Single-cell suspensions ($10^6$/mL) obtained by mechanical cell dissociation of MLNs collected at day 10 after TNBS or DSS administrations were incubated in complete DMEM medium in the absence or presence of phorbol myristate acetate (PMA; 10 ng/mL, Sigma) and concanavalin A (2.5 μg/mL, Sigma). Cell proliferation was evaluated after 96 h by thymidine incorporation as described above. Cytokine production in culture supernatants was determined after 48 h of culture by using specific sandwich ELISAs (BD Pharmingen) as described above. To measure the immunosuppressive activity of analogue 5 in vitro, MLNs cells ($10^5$) isolated from TNBS-induced colitic mice were cultured with analogue 5 (100 nM) and stimulated with PMA plus concanavalin A as described above.

**Flow cytometric analysis.** To determine the number of cells expressing cytokines, single-cell suspensions isolated from MLNs 10 days after TNBS or DSS administration were activated with PMA (25 ng/mL) for 14 h in the presence of monensin (1.33 μM, Sigma) for the last 6 h. Cells were then incubated in 100 μL with anti-mouse CD16/CD32 antibody (clone 2.4G2, 1:100, 10 min at 4 °C, BD Pharmingen, cat #553130) to avoid nonspecific binding to Fc-receptors and with 7-Aminoactinomycin

D (7-ADD, 1:100, Calbiochem) to exclude dead cells. After washing in PBS/0.1% BSA, cells were surface stained with allophycocyanin (APC)-labelled anti-mouse CD4 monoclonal antibody (at 5 μg/mL, for 30 minutes at 4 °C, clone RM4-5, BD Bioscience, cat #: 553051). After extensive washing, cells were fixed/permeabilized with Cytofix/Cytoperm solution (BD Biosciences) and stained with phycoerythrin (PE-CF594)-labelled anti-mouse IL17 (clone TC11-18H10, BD Horizon, cat #: 562542) and FITC-labelled anti-mouse IFNγ (clone XMG1.2, BD Bioscience, cat #: 554411) monoclonal antibodies, or with FITC-labelled anti-mouse IL10 (clone JES5-16E3, BD Biosciences, cat #: 554466) and PE-labelled anti-mouse IL4 (clone 11B11, BD Bioscience, cat #: 554435) monoclonal antibodies (all used at 2 μg/mL, for 30 minutes at 4 °C, in a volume of 100 μL). After extensive washing, samples were acquired and analysed in a FACScalibur flow cytometer (BD Bioscience).

For FoxP3 staining, MLN cells were isolated from mice 10 days after initiating TNBS or DSS administration, treated with anti-CD16/CD32 antibody and 7-AAD as described above, and then cell surface stained with FITC-labelled anti-mouse CD25 (clone 7D4, BD Bioscience, cat #: 553072) and APC-labelled anti-CD4 monoclonal antibodies (both at 5 μg/mL, for 1 h at 4 °C). After extensive washing, cells were fixed/permeabilized (eBioscience), incubated with phycoerythrin-labelled anti-FoxP3 antibody (clone FJK-16S, at 5 μg/mL, eBioscience, cat #: 12-5753-82) for 30 min at 4 °C and analysed in a FACScalibur flow cytometer. In all cases, we used isotype-matched antibodies (BD Biosciences) as controls. Data were acquired until at least 50,000 events were collected from a live gate using forward/side scatter plots and 7-ADD staining (gating and acquisition strategy is described in Supplementary Fig. 6f, g).

**Determination of gene expression by real-time PCR**. Total RNA was isolated from colon sections obtained at day 10 after TNBS injection following the manufacturer's protocol (Tripure, Roche). Precipitated RNA was treated with DNase 1 (Sigma) before reverse transcription (RevertAid First Strand cDNA Synthesis Kit, ThermoFisher Scientific). SYBER green quantitative PCR (Sensi-Fast Sybr No-Rox mix, Bioline) was performed on the Bio-Rad CFX using the following conditions: 95 °C for 5 min followed by 40 cycles at 95 °C for 30 s, annealing at 60 °C for 35 s and extension at 72 °C for 30 s. Primer sequences are included in Supplementary Table 4. The $2^{-\Delta\Delta CT}$ method was used to determine the relative mRNA levels.

**Data analysis**. All values are expressed as mean ± SEM of mice/experiment. For statistical analysis, Prism GraphPad 7 software was utilised. The differences between groups (single comparison between untreated and treated animals/cultures) were analysed by the non-parametric two-tailed Mann–Whitney $U$ test (for analysis of disease index, colon damage, colitis and microscopic scores) or by the paired or unpaired, two-tailed Student's $t$ test (for analysis of proliferation, cytokine levels, changes in body weight and flow cytometry). Survival curves were analysed by the Kaplan–Meier log-rank test.

**Ethic statement**. The experiments reported in this study were approved by the Animal Care and Ethical Committee of Spanish Council of Scientific Research (CSIC) and performed in compliance with the guidelines from Directive 2010/63/EU of the European Parliament on the protection of animals used for scientific purposes. Balb/c and C57Bl/6 mice were purchased from Charles River and were housed in the specific pathogen-free (SPF) animal facility of IPBLN-CSIC in a controlled-temperature/humidity environment (22 ± 1 °C, 60–70% relative humidity) in individual cages (8–10 mice per cage, with wood shaving bedding and nesting material), with a 12-h light/dark cycle (lights on at 07:00 h) and were fed rodent chow (Global Diet 2018, Harlan) and tap water ad libitum. Mice were allowed to acclimatise to their housing environment for at least 5 days prior to experimentation and to the experimental room for 1 h before experiments. Animal studies are reported in compliance with the ARRIVE guidelines. Carbon dioxide inhalation was used for euthanasia. Additional information on experimental design is included in the Nature Reporting Summary file linked to this article.

**Reporting summary**. Further information on research design is available in the Nature Research Reporting Summary linked to this article.

## Data availability
All relevant data are available from the authors upon request. Accession codes for A5 are: PDB: 6Y1Q and BMRB: 34491. Source data are provided with this paper.

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

## Acknowledgements

A.Rol was a recipient of a PhD fellowship from the Generalitat de Catalunya (FI) and A. E. and E.P. were recipients of PhD fellowships granted by the Severo Ochoa Program (FPI). T.T. was a postdoctoral fellow co-funded by the Marie Skłodowska-Curie COFUND actions (IRB Barcelona Interdisciplinary Postdoc Programme). This work was supported by the following grants: CTQ2014-56361-P and CTQ2017-87840-P (A.Riera) and RTI2018-100700-B-100 (M.D.) from the Spanish Ministry of Economy, Industry and Competitiveness (MINECO); and by AGAUR (SGR-50). We also acknowledge institutional funding from MINECO through the Centers of Excellence Severo Ochoa Award given to IRB Barcelona, as well as from the CERCA Program of the Generalitat de Catalunya. M.J.M. is an ICREA Programme Investigator.

## Author contributions

M.D., M.J.M, B.P. and A.Riera designed and supervised the project. A.Rol, P.M-M., E.A. and M.J.M. assigned and analysed the NMR data, and performed NMR measurements and computational analysis. A.Rol, A.E., T.T. and M.V-M synthesised the peptides. J.F-C, X.V. and J.F-S supervised the synthesis. E.G-R and M.D. performed the mouse experiments. E.P. assisted the first and senior authors with manuscript coordination. P.M-M and M.J.M. analysed the binding profiles. M.J.M. and M.D. wrote the paper with input from P.M-M., J.F-C, B.P., and A.Riera. All the authors contributed ideas to the project.

## Competing interests

Analogue 5 is patented, EP 3046933 B1 (BCN PEPTIDES, S.A.) 2019-02-27, "Cortistatin analogues for the treatment of inflammatory and/or immune diseases". All authors declare no other competing interests.
