## [Peer Review File · Nature Communications]

Reviewers' comments:

Reviewer #1 (Remarks to the Author):

see below

Reviewer #2 (Remarks to the Author):

The manuscript by Rol et al contains data relating to the structure and binding properties of analogues of corticostatin as well as studies of the capacity of one such analogue to prevent inflammation in two murine models of IBD, the DSS-colitis and the TNBS-colitis model. The data presented are clear and convincing; however, they offer no new insights into the cause of IBD and are not novel with respect to IBD treatment as it is known from previous studies that experimental colitis is ameliorated by treatment with native corticostatin.

Specific Comments:

1. In only one study was the A5 analogue compared with native corticostatin with respect to efficacy in treating experimental DSS-colitis. In this study the native compound was at least as efficacious as the analogue (A5). Thus, whereas the analogue may have a longer half-life than the native compound, it is not clear that it is a superior treatment modality.
2. Whereas the data provided clearly establishes the efficacy of the A5 analogue, no data is provided on the effect of treatment on secretion of major pro-inflammatory cytokines such as TNF α , IL-1b, IL-6, IFN γ and IL-17. It would be interesting if the analogue had unique effects in this respect.
3. Although data is provided on effects of analogue 5, anti-TNF α and mesalamine on experimental colitis, no "head-to-head" comparisons are presented in which analogue 5 effects is compared with other agents in a single study. This is important to make the case that the analogue is a possible replacement of more conventional therapy.

Based on a structure-function relationship Rol et al. demonstrates the finding of a cortistatin analog that appears to be highly effective in animal models of inflammatory bowel disease. The starting point of the study is the 3D NMR structure of cortistatin in aqueous solution showing multiple conformational states. Changes in the peptide composition resulted in a peptide that was not only longer available (ca 10 fold) in blood plasma but also converged into a single structure with a distinct hydrophobic patch comprising 3 aromatic rings and somatostatin receptor selectivity for receptor SSTR3 and SSTR5. This is an interesting study. However, there are several important aspects to be resolved before publication can be recommended.

Major points:

- (i) The presented structure of analog 5 with the three aromatic rings on one side appears to be unique. It is suggested to superimpose the various structures found with the NMR-derived SSTR1-5 pharmacophores defined by Rivier et al and correlate them with the receptor selectivities observed.
- (ii) Furthermore, it is noted that the orientation of the hydrophobic cluster in respect to the Trp/Lys pair important for receptor activation is switched from cortistatin to analog5 (top in Figure 1 to bottom in Figure 5) because of a backbone reconfiguration of the N-terminal residues. This difference should not only be discussed, but also suggests that the conformation of analog5 is not present in cortistatin, which is at odds to the activities shown. This apparent discrepancy should be resolved.
- (iii) The NMR measurements are not described well (concentration and buffer of the NMR samples should be given). Scalar coupling measurements may be useful for defining the Chi1 aromatic angles. The reviewer indicated some peaks in the Suppl. Figure below that show no assignment that are requesting some clarification.
- (iv) Furthermore, how can Thr QG11 see the aromatic ring of residue6 and residue5 and 7(NOESY in Fig. 5). All of them appear to be rather far in distance. Could it be that there is a lot of spin diffusion attributed to the long mixing time used (i.e. 350 ms). If so, distances can not be determined and shorter mixing times must be used. Alternatively, the sample may oligomerize. Measurements with lower concentrations are thus suggested.
- (v) Why have Lys3 and Lys14 the gammas at such high field (i.e. low ppm) in analog 5?

Supplementary Figure 5. Amide/proton alpha (left) and amide/aromatic (right) regions of the analog **5** (^1H -Noesy experiment), with peak assignments.

Minor points:

- (i) It is suggested that, the NMR structure is represented by 10 lowest conformers (and not structures) using the standard NMR nomenclature.
- (ii) It is requested to deposit the coordinates to the PDB data base
- (iii) In Figure 6b the backbone is only shown, but the labeling does not fit the backbone trace.

Manuscript reference: NCOMMS-19-38507

Summary of revisions

We would like to thank the reviewers and for their insightful comments and recommendations. As a result of these reviews, we have substantially revised and improved our manuscript. The additions include a comparison of the new pharmacophore to other previously described in the literature, new data on the effect of treatment on the secretion of major pro-inflammatory cytokines, and the analysis of histopathological signs of inflammatory infiltration and colonic damage in all the experimental groups, to correlate the clinical signatures as well as macroscopic damage with microscopic scores.

We have also revised the relevant **Results and Discussion sections** to reflect these changes and to emphasize the novelty of our results: the design of a new Cortistatin analog that recapitulates specific structural characteristics of the native hormone and all its anti-inflammatory properties with improved stability. In fact, native Cortistatin has a half-life limited to minutes and even though this short time might not represent an issue in assays with small animal models, it can compromise its efficient distribution and curative effects when administered to larger species like humans.

This new version contains **seven new figure panels** presenting **new data** as well as all suggested figure modifications to improve the representation of previous results.

Reviewers' comments:

Reviewer #1 (Remarks to the Author):

General comments:

Based on a structure-function relationship *Rol et al.* demonstrates the finding of a Cortistatin analog that appears to be highly effective in animal models of inflammatory bowel disease. The starting point of the study is the 3D NMR structure of Cortistatin in aqueous solution showing multiple conformational states. Changes in the peptide composition resulted in a peptide that was not only longer available (ca 10-fold) in blood plasma but also converged into a single structure with a distinct hydrophobic patch comprising 3 aromatic rings and somatostatin receptor selectivity for receptor SSTR3 and SSTR5. This is an interesting study. However, there are several important aspects to be resolved before publication can be recommended.

Major points:

(i)The presented structure of analog 5 with the three aromatic rings on one side appears to be unique. It is suggested to superimpose the various structures found with the NMR-derived SSTR1-5 pharmacophores defined by *Rivier et al* and correlate them with the receptor selectivities observed.

Reply: We thank the reviewer for this recommendation to highlight the novel characteristics of this analog. Thus, we have compared the aromatic packing observed in Analog 5 to other analogs (and pharmacophores) described in the literature as representative models of SSTR binders. To facilitate the comparison and since PDB entries are not available for most of these analogs, we have compared the Phi, Psi and Chi1 values and represented them using conserved residues (**Supplementary Figure7c**). These comparisons reveal that most analogs are very similar with respect to the main features of the backbone conformation in solution, specifically for residues surrounding the Trp-Lys area. We have also analyzed the distances between aromatic residues in Analog 5 and compared them to values reported in the literature and to the Cortistatin conformers determined in solution (**new panel 7e**). According to this analysis, the aromatic cluster observed in Analog 5 is highly unique, defined by the packing of three aromatic residues on one side of the backbone plane, in addition to the classic Trp-Lys

beta hairpin. The presence of these two clusters seems to favor the interaction with receptors SSTR3, SSTR4 and SSTR5 (this text is included in page 12 of the revised manuscript).

(ii) Furthermore, it is noted that the orientation of the hydrophobic cluster in respect to the Trp/Lys pair important for receptor activation is switched from Cortistatin to analog5 (top in Figure 2 to bottom in **new Figure 7**) because of a backbone reconfiguration of the N-terminal residues. This difference should not only be discussed, but also suggests that the conformation of analog5 is not present in Cortistatin, which is at odds to the activities shown. This apparent discrepancy should be resolved.

Reply: We apologize for this confusion, as it seems that our representation of the Cortistatin conformers (**Figure 2d**) failed to illustrate the similarities with the aromatic packing observed in Analog 5. To better clarify these similarities, we have represented the comparison of Analog 5 with Cortistatin as a side by side view, (**new Figure 7c**, left and right panels, with both aromatic clusters labeled). We have also included a second orientation of Analog 5 conformers, (rotated by 180 degrees with respect to the original figure, **new Figure 7d**, right panel). We believe that these new representations facilitate the comparisons.

(iii) The NMR measurements are not described well (concentration and buffer of the NMR samples should be given). Scalar coupling measurements may be useful for defining the Chi1 aromatic angles. The reviewer indicated some peaks in the Suppl. Figure below that show no assignment that are requesting some clarification.

Reply: Thank you for noticing that we forgot to include the solvent and the final concentration of the samples. For each compound, we have prepared 0.5 ± 0.1 mM solutions, by adding 500 microL of pure water and 50 microL of D₂O to the lyophilized product (pH adjusted to be 5.5 with HCl and/or NaOH). This information is now included in the revised version of the material and methods section. Unfortunately, we could not determine the Chi1 angles for AMX systems of the aromatic residues due to the proton overlapping of the corresponding CH₂ groups (as described in the chemical shift table).

We have also included the assignments requested in the **Supplementary Figure 7a**.

(iv) Furthermore, how can Thr QG11 see the aromatic ring of residue6 and residue5 and 7 (NOESY in Fig. 5). All of them appear to be rather far in distance. Could it be that there is a lot of spin diffusion attributed to the long mixing time used (i.e. 350 ms). If so, distances cannot be determined and shorter mixing times must be used. Alternatively, the sample may oligomerize. Measurements with lower concentrations are thus suggested.

Reply: The chemical shifts of Thr9 and Thr11 (Methyl groups) are nearly overlapped but the confusion was due to a mislabeled resonance corresponding to QD10 and QE5. This feature was not properly indicated in the previous version of the figure (now corrected). The NOEs that the reviewer meant correspond to Thr9 (now labeled in **new Fig7c and 7d**) with the aromatic rings of Phe10 and Phe6. All these distances are satisfied in the conformers displayed in the figure (within a 20% error limit used in the volume integration).

We did not observe aggregation of this analog at this concentration. We have acquired NOESY experiments at several mixing times to evaluate the contribution of spin diffusion for these analogs and found that 350ms was a good compromise to study the conformational equilibrium of these peptides in solution.

We have also acquired a dilution set of 1Ds experiments in H₂O to illustrate the lack of aggregation at this range of concentrations (**new Supplementary Figure 7b**).

(v) Why have Lys3 and Lys14 the gammas at such high field (i.e. low ppm) in analog 5?

Reply: Both Lys have gamma proton resonances in the chemical shift characteristic of Lys residues according to the data collected at the BMRB (1.36 ± 0.27 ppm, values taken from http://www.bmrb.wisc.edu/ref_info/statful.htm).

Minor points:

- (i) It is suggested that, the NMR structure is represented by 10 lowest conformers (and not structures) using the standard NMR nomenclature.

Reply: We have corrected this term in the text.

(ii) It is requested to deposit the coordinates to the PDB data base.

Reply: Assignments and coordinates have been deposited at the PDB and BMRB databases for Analog5, with accession codes 6Y1Q and 34491 respectively.

(iii) In Figure 6b the backbone is only shown, but the labeling does not fit the backbone trace.

Reply: The labeling has been corrected. Thank you.

Reviewer #2 (Remarks to the Author):

The manuscript by Rol *et al* contains data relating to the structure and binding properties of analogues of cortistatin as well as studies of the capacity of one such analogue to prevent inflammation in two murine models of IBD, the DSS-colitis and the TNBS-colitis model. The data presented are clear and convincing; however, they offer no new insights into the cause of IBD and are not novel with respect to IBD treatment as it is known from previous studies that experimental colitis is ameliorated by treatment with native corticostatin.

Reply to general comments:

IBD is the result of multiple factors (diet, genetic alterations, environmental and/or microbial inputs) that dysregulate the gastrointestinal immune system and give rise to inflammation and other symptoms collectively known as Inflammatory Bowel Disease (IBD). The underlying mechanisms involved in IBD are not fully understood, although it is accepted that these mechanisms correlate with elevated levels of pro-inflammatory cytokines, tumor necrosis factor- α (TNF- α), interferon-gamma (IFN γ) as well as with interleukin 2 (IL-2).

Our work was aimed at identifying new molecules that help to treat this disease and we specifically focused on the pro-inflammatory mechanisms of the disease. With this in mind, we considered the anti-inflammatory capacity of native Cortistatin as the hallmark to design new analogs that retain its pharmacological potential but with increased stability, therein overcoming the limitations of the short half-life displayed by the native hormone.

To assist the rational design of the mutations, we first studied the conformations of the native hormone in solution by nuclear magnetic resonance (NMR). Using this structural knowledge, we designed the Analog 5 sequence, which showed an increased stability in serum as well as maintaining the extraordinary anti-inflammatory capacity of native Cortistatin. Moreover, this analog can be easily prepared in large scale and at an affordable cost, representing a benefit when compared to other molecules in the market. These advantages position Analog 5 as a strong candidate to treat IBD patients, specially to those that fail to respond to other therapies.

Specific Comments:

- 1. In only one study was the A5 analogue compared with native corticostatin with respect to efficacy in treating experimental DSS-colitis. In this study the native compound was at least as efficacious as the analogue (A5). Thus, whereas the analogue may have a longer half-life than the native compound, it is not clear that it is a superior treatment modality.*
- 2. Whereas the data provided clearly establishes the efficacy of the A5 analogue, no data is provided on the effect of treatment on secretion of major pro-inflammatory cytokines such as TNFa, IL-1b, IL-6, IFNg and IL-17. It would be interesting if the analogue had unique effects in this respect.*
- 3. Although data is provided on effects of analogue 5, anti-TNFa and mesalamine on experimental colitis, no "head-to-head" comparisons are presented in which analogue 5 effects is compared with other agents in a single study. This is important to make the case that the analogue is a possible replacement of more conventional therapy.*

A point-by-point reply to the specific comments raised by the referees follows.

Replies to specific comments:

Q1. In only one study was the A5 analogue compared with native *cortistatin* with respect to efficacy in treating experimental DSS-colitis. In this study the native compound was at least as efficacious as the analogue (A5). Thus, whereas the analogue may have a longer half-life than the native compound, it is not clear that it is a superior treatment modality.

Reply: Natural endogenous peptide hormones can display medical properties but their pharmacological applications are often a challenge due to rapid proteolysis in plasma. In the case of Cortistatin, its half-life is limited to minutes and even though this time might not represent an issue in assays with small animal models, it can compromise its efficient distribution and curative effects when administered to larger species like humans.

In addition, its low stability poses a challenge to its storage in solution, as well as to the administration mode itself, since the solution has to be prepared right before the injection in order to maintain its therapeutic effects. These limitations, which are common for most endogenous peptides, were overcome in this work by introducing modified amino acids to the native sequence, yielding Analog 5, which retains the extraordinary anti-inflammatory capacity of Cortistatin, but with increased stability.

This analogue is also as efficient as the anti-TNF antibody currently used in treatments. Such remarkable features open the door for the use of Cortistatin Analogs for pharmaceutical applications, thus expanding the repertoire of drugs that can help to treat patients (complementing or as an alternative to the ones currently in the market). We cannot forget the economic benefit that these analogs might represent to the public health system since the costs of producing a peptide (Analog 5) cannot be compared to those required to produce and store other molecules or biologics currently found in the market. We have included some of these explanations in the modified discussion (highlighted in blue).

Q2. Whereas the data provided clearly establishes the efficacy of the A5 analogue, no data is provided on the effect of treatment on secretion of major pro-inflammatory cytokines such as TNF α , IL-1b, IL-6, IFN-g and IL-17. It would be interesting if the analogue had unique effects in this respect.

Reply: We agree with reviewer that determination of major immune mediators of IBD helps illustrate the mechanisms involved in the therapeutic effect of analog 5. We have followed this advice and determined the levels of various inflammatory, Th1 and Th17 cytokines (TNF α , IL1b, IL6, MIP2, IL17) as well as the anti-inflammatory cytokine IL10 in the colonic protein extracts and serum in both DSS and TNBS models.

These effects have been compared to the treatments with anti-TNF antibody and mesalazine used as controls as well as with Cortistatin itself (values obtained in this work and in the literature). Results are depicted in **new Figure 6**, and demonstrate that treatment with analog 5 significantly decreases this panel of inflammatory mediators, while increases the amounts of IL10 in colonic mucosa. When these effects are compared to the controls anti-TNF α and mesalazine, Analog 5 displays a unique effect with respect to the amounts of IL10 probably supporting the tolerance effect induced by our analog versus the controls.

To further support our results, we have analysed histopathological signs of inflammatory infiltration and colonic damage in all the experimental groups, in order to correlate the clinical signatures as well as macroscopic damage with microscopic scores. We have included the histopathological analysis in **new Figures 3 and 4, and Supplementary Figure 5**. The sections describing these figures have also been revised.

Q3. Although data is provided on effects of analogue 5, anti-TNF α and mesalamine on experimental colitis, no “head-to-head” comparisons are presented in which analogue 5 effects is compared with other agents in a single study. This is important to make the case that the analogue is a possible replacement of more conventional therapy.

Reply: Thank you for this recommendation.

We performed the experiments as a single study. Please, notice that DSS and TNBS controls are the same in all figures, (main and supplementary) indicating that all experiments were performed in parallel. However, we thought that due to the complexity of the figures, we represented the data as separate figures. We agree with the reviewer that the previous figures might have not been the best option to illustrate our results. Following these recommendations, we have displayed the effects of analog 5, mesalazine and anti-TNF α antibody “head-to-head” (**new Figure 3, Figure 4 and Figure 5**). In addition, the new histopathological data and cytokine determinations were also represented next to the experiments, to facilitate the analysis.

REVIEWER COMMENTS

Reviewer #1 (Remarks to the Author):

The revised manuscript by Rol et al on a new SST14 construct that binds to SSTR3/SSTR5 selectively and has interesting features towards a drug against inflammatory bowel disease is much improved including implementations requested from the reviewer. The now implemented pharmacophore comparison does beautifully show the uniqueness of analog 5. It would therein be interesting to note that the published pharmacophore shown for SSTR2/SSTR5 includes the analog 5 pharmacophore.

There are the following remaining questions and requests of minor level:

- (i) In Figure 7e and Suppl. Figure 7, please rotate the pharmacophores such that they have always the W K pair at the same position.
- (ii) In Table 1, the error bars of the measurements must be added.
- (iii) In Table 2, the error bars of the measurements must be added.
- (iv) Figure caption 7: NOESY should be written with capital letters.

Reviewer #3 (Remarks to the Author):

Rol et al. assessed the therapeutic effect of a newly designed Cortistatin analog in inflammatory bowel disease in different mouse models. First they described the different designed analogs and tested for pharmacological and immune activating properties to identify the most promising structure for further in vivo experiments. Therefore, they used different acute and chronic mouse models using DSS and TNBS as trigger. The authors clearly showed reduced colitis development in Cortistatin analog treated mice in all models. However, no analysis was done to identify the mechanism or to characterize the cellular immune response. This general comment was already raised before and has not been resolved.

Major concerns:

I'm aware that the major aim of this work is to find a potential candidate protein for IBD treatment and the analog 5 seems to be a good candidate; however, the authors provide no insights into relevant pathways or mode of actions. E.g., are regulatory cells induced after analog 5 treatment, or do inflammatory cells undergo apoptosis? This should be analysed by the use of more complex models than already done.

I also acknowledge the use of two different animal models, each in a different mouse strain, and the use of subacute ones. However, none of those is really a chronic model. Anti-inflammatory properties of some agents have resulted in beneficial effects in acute to subacute models but to detrimental effects in chronic ones. A four cycle DSS treatment or the use of genetically modified models is helpful in this respect. The latter could also give more insights into the mode of action. In addition, the authors claim translational value of the findings, which is appropriate. However, the use of just chemical induced models in one species limits the external validity in my eyes.

Table 1: the experiment was performed one time in duplicates. The authors should repeat the experiments to show results of 3 independent runs. Furthermore, the authors might consider elaborating the consequences of the findings in the context of inflammation.

Table 2: Here the authors performed the experiment in three independent experiments but they show only the mean. Please also provide SEM as well as the data in the supplemental data file. Why did the authors use 100nM of the peptides? Did they check the concentration in a preceding test? What do the cells look like after culture? Did the authors observe an induction of a specific T cell population or an increase in apoptotic cells?

Fig. 3-5: Why did the authors change the administration routes of analog 5 and control substances (Cortistatin, TNF, Mesalazine) in these experiments?

I missed the histological slides and data for the acute DSS-colitis model. For determining intestinal

inflammation, I consider the histological analysis as quite important. Therefore, the use of 3 animals per group represents a rather low number, especially as statistical tests were performed to underline group differences, clearly resulting in loss of power.

The authors should adhere to the ARRIVE guidelines when describing in vivo work. Strain names are partly misspelled (C57BL/6), substrain information is not given (N or J?), health status and housing conditions are missing, as do the information on randomization and blinding, to name a few.

Minor points:

Fig.5c the TNBS+anti TNF chronic group is shown without SEM.

The description/definition for tolerance (Line 249) is hardly to understand, please clarify.

The manuscripts contains quite a number of misspellings, this should be corrected.

Point-by-point response to the reviewers' comments

Reviewer #1 (Remarks to the Author):

The revised manuscript by Rol et al on a new SST14 construct that binds to SSTR3/SSTR5 selectively and has interesting features towards a drug against inflammatory bowel disease is much improved including implementations requested from the reviewer. The now implemented pharmacophore comparison does beautifully show the uniqueness of analog 5.

It would therein be interesting to note that the published pharmacophore shown for SSTR2/SSTR5 includes the analog 5 pharmacophore.

Reply: Thank you for your help during the revision of the manuscript. This recommendation is now included in the revised version (page 13).

There are the following remaining questions and requests of minor level:

Query (i). In Figure 7e and Suppl. Figure 7, please rotate the pharmacophores such that they have always the W K pair at the same position.

Reply (I): We have modified the figures to adjust the orientation of the pharmacophores as recommended.

Query (ii). In Table 1, the error bars of the measurements must be added.

Reply (II): We have included the values and error bars corresponding to the IC₅₀ experiments performed. These new changes are reflected in the new Figure 1d, Supplementary Figure 1b and Supplementary Table 1.

Query (iii). In Table 2, the error bars of the measurements must be added.

Query (iv) Figure caption 7: NOESY should be written with capital letters.

Reply (III and IV): They have now been included.

Reviewer #3 (Remarks to the Author):

Rol et al. assessed the therapeutic effect of a newly designed Cortistatin analog in inflammatory bowel disease in different mouse models. First they described the different designed analogs and tested for pharmacological and immune activating properties to identify the most promising structure for further in vivo experiments. Therefore, they used different acute and chronic mouse models using DSS and TNBS as trigger. The authors clearly showed reduced colitis development in Cortistatin analog treated mice in all models. However, no analysis was done to identify the mechanism or to characterize the cellular immune response. This general comment was already raised before and has not been resolved.

Major concerns:

Query 1: I'm aware that the major aim of this work is to find a potential candidate protein for IBD treatment and the analog 5 seems to be a good candidate; however, the authors provide no insights into relevant pathways or mode of actions. E.g., are regulatory cells induced after analog 5 treatment, or do inflammatory cells undergo apoptosis? This should be analysed by the use of more complex models than already done.

I also acknowledge the use of two different animal models, each in a different mouse strain, and the use of subacute ones. However, none of those is really a chronic model. Anti-inflammatory properties of some agents have resulted in beneficial effects in acute to subacute models but to detrimental effects in chronic ones. A four cycle DSS treatment or the use of genetically modified models is helpful in this respect. The latter could also give more insights into the mode of action. In addition, the authors claim translational value of the findings, which is appropriate. However, the use of just chemical induced models in one species limits the external validity in my eyes.

Reply 1: The main aim of this study was to demonstrate a new strategy of designing stable analogs based on the structure of a well-documented immunoregulatory hormone, Cortistatin, and using various *in vitro* studies and two established preclinical models of IBD as a proof of concept of their potential therapeutic effects, in comparison with treatments of reference already used in the clinic. Our initial goal was not to provide an in-depth description of mechanisms involved in the therapeutic effect of this stable analog. However, we agree with this reviewer that better identification of the immune mechanisms would contribute to understanding its mode of action and facilitate the design of other analogs in the future. Thus, following the reviewer's recommendation, several additional experiments have been performed to **characterize the mechanism of action** of Analog 5 in comparison with Cortistatin:

- **Th2 responses in vitro.** Treatment with Cortistatin and/or Analog 5 reduced the production of IL4, IL5 and IL13 by OVA-stimulated spleen cells isolated from mice immunized with OVA (shown in the new Figure 2c). These findings indicate that, in addition to inhibiting inflammatory and Th1 responses in macrophages and lymphocytes, Analog 5 is also able to impair Th2-driven responses.
- **Apoptosis of activated macrophages and lymphocytes in vitro.** Analog 5 did not affect the viability or apoptosis of these cells (Supplementary Table 2), indicating that most probably this mechanism does not contribute to the observed immunoregulatory actions.
- **Mesenteric lymph node isolated from DSS- and TNBS-colitis mice.** We found that treatment with Analog 5 resulted in MLN cells that proliferate less and express lower levels of Th1 cytokines and higher levels of IL10 upon ex vivo re-stimulation. These data are shown in the new Figure 6 and Supplementary Figure 6. Moreover, flow cytometric analysis of MLNs showed that treatment with Analog 5 decreased the presence of Th17 and Th17 cells and increased the percentage of IL10-producing CD4 T cells and CD25+FoxP3+ CD4 T cells. Moreover, addition of Analog 5 to MLN cultures from colitic mice deactivated the responses of Th1 and increased those of IL10. These data in draining lymph nodes of colon correlate with our previous observations on cytokine levels in colonic mucosa. In addition, we have found that treatment with Analog 5 downregulates serum levels of antibodies against TNBS, probably as a consequence of an indirect regulation of

B cells by Th-driven responses. Finally, we found that treatment with Analog 5 reduced the gene expression of ROR-gt and T-bet and increased the expression of FoxP3 in the colon of colitic mice, thereby indicating that this treatment impairs the presence of Th1 and Th17 and increases the presence of Treg cells in inflamed colonic mucosa. All these new experiments demonstrate that Analog 5 can directly limit the inflammatory and Th1 and Th17 responses at the local level in colonic mucosa and impair the activation and/or differentiation of Th1 and Th17 cells and induce IL10-producing lymphocytes and Treg cells in peripheral lymphoid organs. These effects could be related to the response of tolerance or partial resistance observed after an initial treatment with the analog and posterior re-induction of colon damage.

Our strategy of using several established models of IBD was based on the fact that both DSS and TNBS models have been widely used in the literature for assaying new therapeutic agents (validated by clinical treatments of reference such as anti-TNF antibody and mesalazine) (see references below). We used acute models as a proof of concept. The injection of increasing doses of TNBS over four weeks is considered a chronic model of IBD. DSS in two cycles over four weeks was used as a relapsing-remitting model, not as a chronic model. We agree with the reviewer that a model with four DSS cycles would reflect a longer chronic model. However, the Ethics Committee for Animal Research of our institution imposed some limitations and did not recommend a study with chronic models, especially of those with high mortality like the DSS model, but rather the use of alternative chronic models such as TNBS.

- Kiesler P, Fuss IJ, and Strober W. Experimental Models of Inflammatory Bowel Diseases. *Cell Mol Gastroenterol Hepatol* 2015;1:154–170.
- Silva I, Pinto R and Mateus V. Preclinical Study in Vivo for New Pharmacological Approaches in Inflammatory Bowel Disease: A Systematic Review of Chronic Model of TNBS-Induced Colitis. *J. Clin. Med.* 2019, 8:1574.

We showed that Analog 5 has a significant effect on the models we used, which cover acute, relapsing-remitting and chronic profiles, in both protective and curative regimes. These effects match current treatments of reference or are even superior. In our opinion, the results provide a solid base for further preclinical studies to develop alternative or complementary molecules for patient treatment. Moreover, if this molecule is successful in preclinical studies, it will be commercialized at a more affordable price than similar treatments currently on the market.

Query 2: Table 1: the experiment was performed one time in duplicates. The authors should repeat the experiments to show results of 3 independent runs. Furthermore, the authors might consider elaborating the consequences of the findings in the context of inflammation.

Reply 2: These experiments were performed as a qualitative comparison of the analogs, since we observed different behaviour in cells (activity of Analog 5 versus non-activity of the remaining analogs). Moreover, for each analog, the IC₅₀ experiments were run with Somatostatin as a positive control. These controls were highly reproducible and in agreement with values reported in the literature. Since the aim of this work was not to design new analogs that could compete with Somatostatin and/or its commercial analogs, we validated the differences observed in IC₅₀ profiles using NMR and not by repeating the experiments in triplicates. Our results showed that the ensemble of conformations populated by Analog 5 differs from the other analogs (Figures 7 and 8), thereby illustrating the distinct properties of this Analog 5 in cells.

We did not pursue the analysis of these analogs to monitor the expression of Somatostatin receptors in models of inflammation, as described in the literature using radiolabel analogs of Somatostatin. We agree with the reviewer that exploring the application of Analog 3 and Analog 5 to classify inflammatory cells on the basis of the expression level of Somatostatin receptors could be another potential application of these molecules. However, we consider that this application requires further research and a more detail analysis of the binding profiles of these two analogs, both with purified receptors and in cells. Therefore, we prefer to avoid premature speculations regarding their potential applications in the context of Somatostatin receptors and inflammation.

Query 3. Table 2: Here the authors performed the experiment in three independent experiments but they show only the mean. Please also provide SEM as well as the data in the supplemental data file. Why did the authors use 100nM of the peptides? Did they check the concentration in a preceding test? What do the cells look like after culture? Did the authors observe an induction of a specific T cell population or an increase in apoptotic cells?

Reply 3: Data on inflammatory and Th1 immune responses by activated Raw macrophages and spleen cells previously showed in Table 2 have now been included in two new figures (Figure 2a,b and Supplementary Figure 2c,d), showing data as mean \pm SEM. We have also included a dose-response curve of Cortistatin and Analog 5 for activated primary peritoneal macrophages and spleen cells (new Supplementary Figure 2a,b), showing that doses between 100 and 500 nM exert the maximal anti-inflammatory and immune effects. The doses we used were based on the observation that both Cortistatin and Analog 5 showed similar downregulation of the inflammatory response and immune activation at similar concentrations.

Following the suggestion of this reviewer to extend the knowledge on the cellular mechanisms involved in the effect of Analog 5, Figure 2 (panel c) also includes a new experiment describing the effects of Cortistatin and Analog 5 in Th2 responses by ovalbumin-sensitized spleen cells. These experiments show that Cortistatin and Analog 5 inhibit Th2 responses in a similar way.

Finally, we have evaluated the effect of Analog 5 on the viability and apoptosis of activated macrophages and spleen cells. We found that Analog 5 does not affect the viability or apoptosis of these cell populations (new Supplementary Table 1), thereby suggesting that programmed cell death is not a potential mechanism involved in the action of Analog 5 in immune responses.

Query 4. Fig. 3-5: Why did the authors change the administration routes of analog 5 and control substances (Cortistatin, TNF, Mesalazine) in these experiments?

Reply 4: We used different routes of administration for Analog 5 in order to select the most effective pathway of administration to treat experimental IBD. We found that systemic treatment by subcutaneous and intraperitoneal injections showed similar efficiency in reducing clinical signs of colitis. Although oral administration of Analog 5 also showed a slight effect in reducing colitis, this effect was not significant enough to continue the study of this route of administration as the one of choice.

Probably, as in the case of other peptide-based therapies, subcutaneous injection will be the most plausible route of administration. Moreover, anti-TNF antibody and mesalazine were used as treatments of reference, and we used the same route of administration and doses/kg used in the clinic for treating IBD patients. All reported studies using Cortistatin in different inflammatory and autoimmune models have used intraperitoneal injections as the preferential route of administration, and we decided to use the same strategy. Because intraperitoneal and subcutaneous administration of Analog 5 showed similar effects on colitis, we decided to use subcutaneous injection in most of the assays as a more plausible route of administration in patients.

Query 5. I missed the histological slides and data for the acute DSS-colitis model. For determining intestinal inflammation, I consider the histological analysis as quite important. Therefore, the use of 3 animals per group represents a rather low number, especially as statistical tests were performed to underline group differences, clearly resulting in loss of power.

Reply 5: We have included histopathological analysis for the acute DSS-colitis model in a new Figure 3a. With respect to the number of animals, we have reduced the number to a minimum following the recommendations of the guidelines of the Ethics Committee of our institution, especially in these animal models, which are categorized as severe. Due to the differences at the histological level between experimental groups, we found that 3 or 4 animals were enough to reach statistical differences between them, and we considered that inflicting pain on more animals per group was not justified.

Query 6. The authors should adhere to the ARRIVE guidelines when describing in vivo work. Strain names are partly misspelled (C57BL/6), substrain information is not given (N or J?), health status and housing conditions are missing, as do the information on randomization and blinding, to name a few.

Reply 6: The guidelines and instructions for authors of Nature Communications establish that all this information (randomization, blinding, replication, data exclusion, sample size determination, housing conditions, ARRIVE guidelines,...) must be supplied in a separate document named Nature Research Reporting Summary, which was loaded together the manuscript in the previous submission. We assumed that reviewers had access to this document. We apologize for the misunderstanding.

Minor points:

Fig.5c the TNBS+anti TNF chronic group is shown without SEM.

Reply: We apologize for this mistake. It has been corrected in the revised version of the manuscript.

The description/definition for tolerance (Line 249) is hardly to understand, please clarify.

Reply: We have rephrased the sentence in the new version of the manuscript.

The manuscript contains quite a number of misspellings, this should be corrected.

Reply: Thank you very much for pointing this out. We have revised these misspellings in the new version of the manuscript.

REVIEWERS' COMMENTS

Reviewer #3 (Remarks to the Author):

I thank the authors for adding additional immunological data to clarify the impact of Cortistatin on immune regulation. In my eyes, this substantiates the in vivo findings and strengthens their paradigm. One minor point: The colitis damage score for TNBS treatment ranges from 0 to 10 (see Material and method); however, the axis in Fig. 4b only reaches a score of 8. The authors might consider adapting the ordinates axis.

REVIEWERS' COMMENTS

Reviewer #3 (Remarks to the Author):

I thank the authors for adding additional immunological data to clarify the impact of Cortistatin on immune regulation. In my eyes, this substantiates the in vivo findings and strengthens their paradigm.

One minor point: The colitis damage score for TNBS treatment ranges from 0 to 10 (see Material and method); however, **the axis in Fig. 4b** only reaches a score of 8. The authors might consider adapting the ordinates axis.

Reply: Thank you very much for your comments and suggestions to improve our work. We have corrected the Y axis in Fig.4b.